# Selectivity for food in human ventral visual cortex

Nidhi Jain[1], Aria Wang[2,3], Margaret M. Henderson [2,3], Ruogu Lin[4], Jacob S. Prince [5,6], Michael J. Tarr [2,3,5,7] & Leila Wehbe [2,3,5,7 ✉]

Visual cortex contains regions of selectivity for domains of ecological importance. Food is an evolutionarily critical category whose visual heterogeneity may make the identification of selectivity more challenging. We investigate neural responsiveness to food using natural images combined with large-scale human fMRI. Leveraging the improved sensitivity of modern designs and statistical analyses, we identify two food-selective regions in the ventral visual cortex. Our results are robust across 8 subjects from the Natural Scenes Dataset (NSD), multiple independent image sets and multiple analysis methods. We then test our findings of food selectivity in an fMRI "localizer" using grayscale food images. These independent results confirm the existence of food selectivity in ventral visual cortex and help illuminate why earlier studies may have failed to do so. Our identification of food-selective regions stands alongside prior findings of functional selectivity and adds to our understanding of the organization of knowledge within the human visual system.

[1] Computer Science Department, Carnegie Mellon University, Pittsburgh, PA, USA. [2] Neuroscience Institute, Carnegie Mellon University, Pittsburgh, PA, USA. [3] Machine Learning Department, Carnegie Mellon University, Pittsburgh, PA, USA. [4] Computational Biology Department, Carnegie Mellon University, Pittsburgh, PA, USA. [5] Department of Psychology, Carnegie Mellon University, Pittsburgh, PA, USA. [6] Department of Psychology, Harvard University, Cambridge, MA, USA. [7] These authors jointly supervised this work: Michael J. Tarr, Leila Wehbe. ✉email: lwehbe@cmu.edu

The representation of high-level visual information in the human brain has been marked by the phenomenon of selectivity for visual categories or properties of high ecological importance. Focusing on ventral visual cortex, there are multiple brain regions that show preferential responses to categories such as faces[1,2], bodies[3], places[4], and words[5], and to broad organizational principles such as animacy[6], real-world size[6], and "reach space"[7]. Independent of any particular theory on the origins and specificity of these functional brain regions[8,9], the prevailing view is that the likely role of these regions is to instantiate processes and representations for categories and properties that are highly relevant for common and important day-to-day behaviors. In a similar vein, food is a category that is relevant to evolution—the need to find nourishment is more ancient than social interaction and, arguably, more fundamental to survival. It is therefore surprising that food has not been consistently identified as a visual category for which localized, selective neural responses are observed.

The visual presentation of food images is known to prompt a range of brain responses[10–13], including affective, sensory, and cognitive effects. However, agreement on neuroanatomical locations of food-related activation across studies using food images has been low to moderate[13]. In one meta analysis of relevant studies, only 41% of 17 experiments contributed to food-related clusters in the bilateral fusiform gyrus and left orbitofrontal cortex[13]. Another study of selectivity across a range of proposed categories found no robust selectivity for either fruits or vegetables in occipitotemporal cortex[14]. In the cases where statistically significant responses to food have been observed, they have typically been attributed to increased attention to food images arising from subjects' mental states and/or physiological factors[11,13,15] rather than to visual category representations per se. For example, supporting the idea that it is the value of particular foods that drives responses, Huerta and colleagues[11] performed a meta analysis across 11 studies specifically focused on eating behavior, where they compared high caloric food pictures (e.g., hamburgers, cake, waffles, fries, etc.) to non-food pictures (e.g., rocks, bricks, trees, houses, etc.) and found the most consistent group-average activation in the right fusiform gyrus[11]. In addition, in the study most relevant to our present work, Adamson and Troiani[16] considered the connection between a subject's body mass index (BMI) and neural responses to food in a paradigm that compared 80 food images to an equal number of faces, places, and clocks. Interestingly, independent of any interaction with BMI, they found evidence for left-lateralized food selectivity, overlapping with the fusiform face area (FFA), and interpreted this as an indication that fusiform activation may be driven by motivation and valence factors that are common to both food and faces. This earlier finding of food selectivity in the FFA was further interpreted as a counter-example to the theory that FFA selectivity is a consequence of "expertize"—high proficiency at individuating exemplars within a visually-similar category[9] (in that food images are relatively dissimilar from one another). However, their conclusion was based primarily on group average responses and focused on establishing overlap between food selectivity and the FFA, rather than parsing the fine-grained anatomical relationship between food- and face-selective populations. Thus, while it is known that food images elicit neural responses in a variety of brain regions, including the fusiform gyrus, it is not yet clear whether selectivity for food images is instantiated as a distinct category-selective region within ventral visual cortex.

To address this question, in contrast to prior work, we do not include any physiological variables (e.g., BMI or hunger level) as covariates in our analyses, and we do not restrict our image set to high-calorie, appetizing stimuli. Rather, our study explicitly aimed to identify the brain regions that represent and process the visual properties of food in a more general context; that is, without explicitly or implicitly attempting to recruit circuits involved in reward, motivation, or valence. We present two experiments relying on very different designs. Experiment 1 uses a large-scale, "hypothesis-free" approach in which fMRI data was collected at a massive scale as part of NSD[17], thereby improving our ability to detect effects across *post-hoc* defined conditions. Real-world images, drawn from the the Microsoft COCO dataset[18], were used for both the food and non-food conditions. To preview our most important result, we reliably identify two distinct regions in ventral high-level visual cortex that are preferentially responsive to food images. These two strips surround the Fusiform Face Area (FFA) and are aligned on the anterior to posterior axis. We replicate these regions across subjects while controlling for other aspects of images that are thought to be coded in the ventral visual system, such as image perspective. We also provide exploratory analyses that probe the more fine-grained structure of conceptual representations within food-selective cortex both across and within individuals. Notably, two other studies[19,20] based on the same Natural Scenes Dataset (NSD)[17] we used in Experiment 1, both identified distinct food-selective regions consistent with these results (although relying on somewhat different analysis methods). We will return to these studies in the Discussion.

Experiment 2 validates the finding of food-selective regions in a hypothesis-driven manner by collecting new fMRI data. We designed a visual "food localizer" by adding a food condition to the existing fLoc localizer by Stigliani et al.[21]. As in the other conditions of the fLoc localizer, we composited grayscale food images on scrambled backgrounds. Our analysis identified food-selective regions in each subject, with the location being consistently adjacent to the FFA. The results of Experiment 2 provide direct evidence supporting the hypothesis that food-selective regions in the ventral visual system represent a new domain of category selectivity similar to faces, places, bodies, and words. Our results also directly exclude color and image context from being the major drivers of the visual responses to food observed in Experiment 1. Of particular note, the localization of the food region was consistent across individuals when defined according to a functional landmark (e.g., proximity to the FFA), but when averaging spatially across individuals (e.g., when their brains were aligned), the neuroanatomical overlap of the food region across subjects was less pronounced than other functional ROIs (and closely replicated the results reported in[16], thereby accounting for the differences between the results of Experiment 1 and earlier studies). This leads us to consider the third and fourth factors— the spatial heterogeneity of food-selective regions and the impact such heterogeneity has on traditional localizer designs—as the leading causes for the elusiveness of food selectivity. We release the food localizer code and stimuli as part of this paper.

Naturalistic and hypothesis-driven experimental approaches can be used in a complementary manner that leverages their unique strengths. Here we were able to identify and validate a food-selective region of the human ventral cortex using a naturalistic experiment with complex stimuli to formulate our hypothesis and then use a hypothesis-driven experiment to test that hypothesis. We believe such a combination is a valuable tool in neuroscience that can help advanced the field in the coming years.

From a theoretical standpoint, in that food is incontrovertibly an ecologically critical category, our finding of a food-selective region (confirmed in[19,20]) is consistent with earlier findings of selectivity in the perception of faces, bodies, places, and words. Building on this result, principal component analyses across food-selective voxels provides a finer-grained view into the rich

organization of food-relevant information within visual cortex, possibly reflecting gradients along which food is combined with other ecologically relevant categories.

## Results

### Experiment 1: Large-scale analyses of food representations in a naturalistic setting.

To investigate responsiveness to food in a large-scale natural setting, we used the Natural Scenes Dataset (NSD)[17], which consists of high-resolution fMRI responses to naturalistic scenes. NSD contains fMRI data from 8 screened subjects (S1–S8) who each viewed 9000–10,000 scene images. Of the 70,566 total unique images viewed across subjects, for purposes of consistency we focused on the 1000 images that were shared among subjects (see *Materials and Methods* for more details).

Though COCO images already include labels for many categories, including some types of food, there is important information not captured by these labels, such as whether an image contains human faces. We methodologically relabeled by hand the 1000 images shared across subjects, based on 3 main attributes: location, content, and image perspective. We used the hierarchical structure shown in Fig. 1b (refer to *Methods* for labeling details, and Fig. 1a for examples). Image perspective was included because there is evidence that objects shown at human-reachable distances have a distinct representational signature in the brain[7,22] and food is often viewed at reachable distances.

Using these labeled images, we constructed a standard linear model that expresses brain activity as a combination of the attributes assigned to each image. This model identified voxels that are more responsive to food than other categories, based on a *t*-test comparing the weights for food vs. all other labels (Fig. 1c). Across the cortex, there are several regions showing significantly higher activation for food than non-food categories ($p < 0.05$, false discovery rate (FDR) corrected), including some areas in parietal and frontal cortex, as well as on the ventral surface of the occipital lobe. We focus on ventral visual cortex due to the long history of mapping category-selective responses in this brain region. Across all subjects, we consistently find two food-selective strips in the ventral visual cortex that surround the FFA on the lateral and medial sides. (Fig. 1c shows the count of subjects for whom these contrasts are significant at each MNI voxel (Montreal Neurological Institute coordinate system), and the contrast strength is shown for individual subjects in Fig. 2a and Suppl. Fig. S1a). Note that these identified regions persist even when removing all images with the "reach" (Suppl. Fig. S3) or "zoom" (Suppl. Fig. S4) annotations—demonstrating that food-selective responses are not dependent on food being shown at a particular distance[7].

Since this paper focuses on visual food selectivity, we isolated fusiform food-selective voxels using a mask of the ventral visual cortex based on corresponding ROIs from the HCP atlas[23] (see *Methods*). The resulting "food relevant" voxel masks, which were used for the following analyses, are shown in Fig. 2b and Supplementary Fig. S1b. We then look at which images maximize the activity in those areas, using a completely separate dataset (the non-shared NSD images). Considering only unique images that were viewed by a given subject, Fig. 1d shows the top 10 activating images for the food-selective voxels for that subject. These images overwhelmingly depict food. These images were not used to identify the food regions, and thus reinforce the generality of food selectivity across independent image sets.

Given that food-selective regions appear adjacent to the FFA, we focused on the spatial relationship between food-selective and face-selective populations on the ventral surface. We compared the *t*-statistics for a contrast of food vs. non-food and *t*-statistics from a

contrast of faces vs. non-faces for S1–S8 individually (Fig. 2a and Suppl. Fig. S1a). The faces vs. non-faces contrast reveals a voxel cluster overlapping with the FFA[1,2] (Fig. 2a and Suppl. Fig. S1a). The FFA was localized for each subject through a separate visual category localizer experiment. (The faces vs. non-faces comparison also makes the methodological point that established category-selective regions can be reliably localized in a large-scale event-related design using stimuli embedded in complex, real-world scenes. This generalizes findings from typical localizer designs and decontextualized images[21]). The regions with higher activity for food are spatially distinct from the ones with higher activity for faces. This pattern persists when comparing food or faces to non-face and non-food images only (Suppl. Fig. S5), indicating that the regions that have high activity for food and faces have highly independent or non-overlapping spatial extents. A somewhat similar pattern of results is seen when we consider the magnitudes of these responses as realized in a measure of voxel-wise selectivity, defined as: $\frac{\text{preferred}-\text{non-preferred}}{|\text{preferred}|+|\text{non-preferred}|}$ where the non-preferred baseline activity is the maximum activity related to other categories (for details see caption of Suppl. Fig. S11). As illustrated in Supplementary Fig. S11, as compared to the *t*-statistics shown in Fig. 2a and Supplementary Fig. S1a, voxel-wise selectivity is more variable across subjects. For some subjects, the strength of selectivity for food is the same as that observed for faces, however, for other subjects food selectivity is greater than for faces, and for still other subjects food selectivity is less than for faces. Such variability across subjects may be due to the fact that the max statistic is sensitive to the number of samples. In Experiment 1, different categories within NSD have different frequencies; in contrast, in Experiment 2 category frequencies are balanced.

We further investigated how food representations might be distributed across multiple voxels, using searchlight classification[24] (Suppl. Fig. S2). Training a decoder to classify food vs. other categories revealed that food was decodable across a wide area of the ventral surface. The regions from which food information was decodable are a union of the regions that are high for food vs. all and the regions that are high for faces vs. all. This finding is consistent with the idea that voxels primarily selective for other categories, such as faces, may contain information that distinguishes food from other categories[25].

We have focused on identifying food-selective regions through responses to the shared images and our hand-labeled annotations. For the ~9000 remaining images per subject that were not manually labeled, we can still take advantage of COCO annotations[18] (including specific types of food) to further investigate brain responses to food and validate our findings on an independent set of images. We built an encoding model using the 80 object labels provided by COCO and obtained the resulting voxel-wise weights for food labels. We find that the voxels having the highest weights for several individual food sub-categories (i.e., *cake, sandwich, pizza, and broccoli*) fall within previously identified food-selective regions (weights for S1 in Fig. 3a). Next, we investigated the specific contribution of food images to these voxel responses by comparing two encoding models: one including the 67 non-food COCO labels, and the other including both food and non-food labels. We compared the $R^2$ values of the two models on held-out data (Fig. 3b and Suppl. Fig. S6). Many voxels on the ventral surface show improved prediction performance due to the inclusion of food labels, suggesting that modeling the presence of food beyond other categories was required to accurately predict the voxel responses. These voxels are distributed in roughly the same spatial pattern as the voxels with high-valued weights for individual food categories and our previously identified food regions, further supporting the generality of our results.

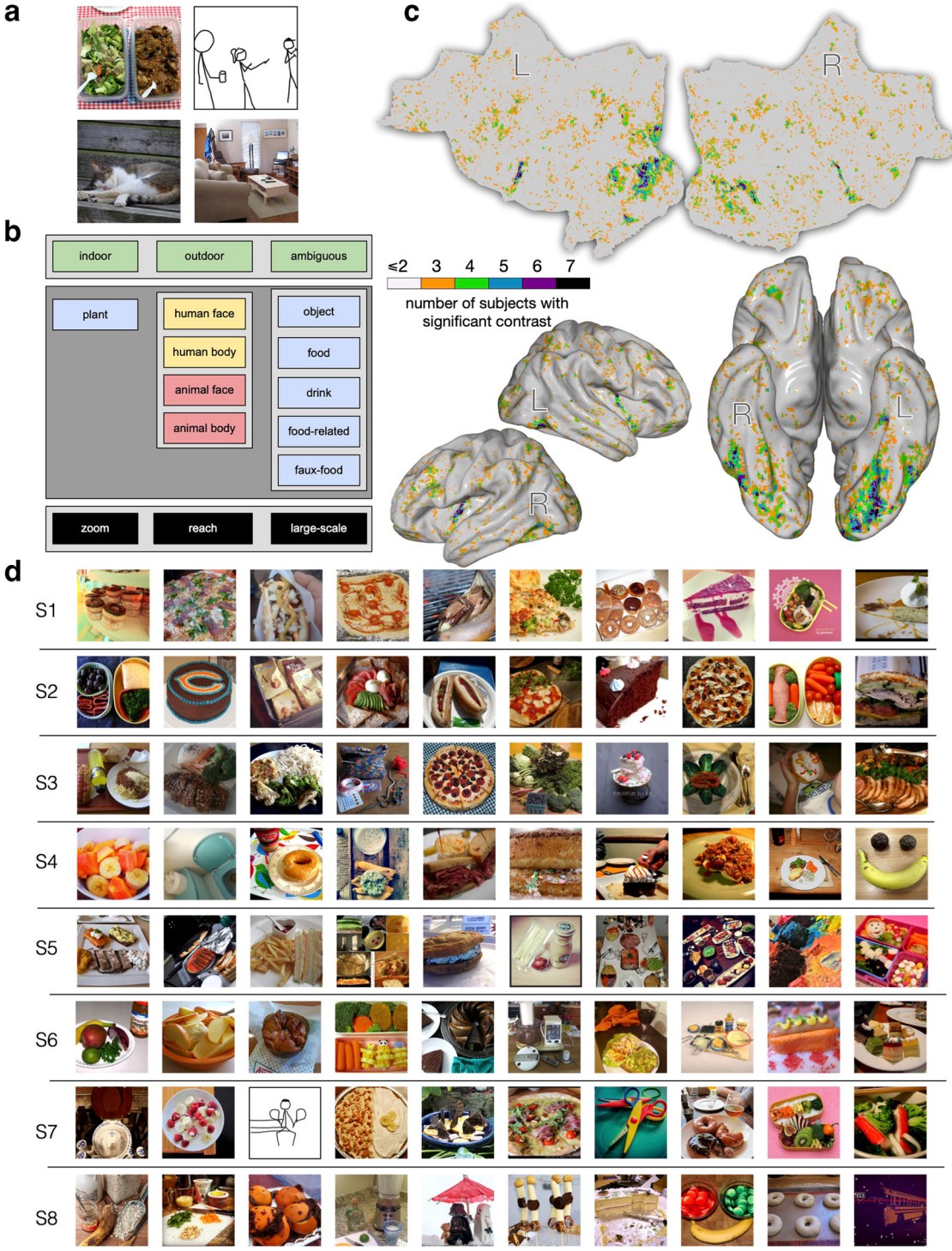

**Fig. 1 Experiment 1. The images that could have been potentially viewed by all 8 subjects in NSD were manually relabeled to investigate responsiveness to naturalistic food images. a** Example images labeled as (clockwise, from upper left): {outdoor, food, food-related, reach} {indoor, human face, human body, object, large-scale}, {indoor, object, large-scale}, {outdoor, animal face, animal body, object, zoom}. **b** The labeling taxonomy, including attributes of location (top), content (middle), and image perspective (bottom). **c** Flattened, semi-inflated lateral, and semi-inflated bottom views of the MNI surface indicating voxels with higher activity for food than all non-food labels for the shared images. The subject count for a significant contrast was obtained at each MNI voxel. Voxels more responsive to food are found in the frontal, insular, and dorsal visual cortex, with the highest concentration across subjects occurring in the fusiform visual cortex. Both hemispheres show two strips within the fusiform that are separated by a gap that lies on the posterior-to-anterior axis. **d** Top 10 images per subject (S1–S8) leading to the largest responses in the food area. These images, which overwhelmingly depict food, were unique for each subject and were not in the set used to localize the food-selective region. Due to licensing concerns, images showing people that were used in our study have been replaced with stick figures representing the structure of the original stimulus images. Replacement images (badly) drawn by MJT.

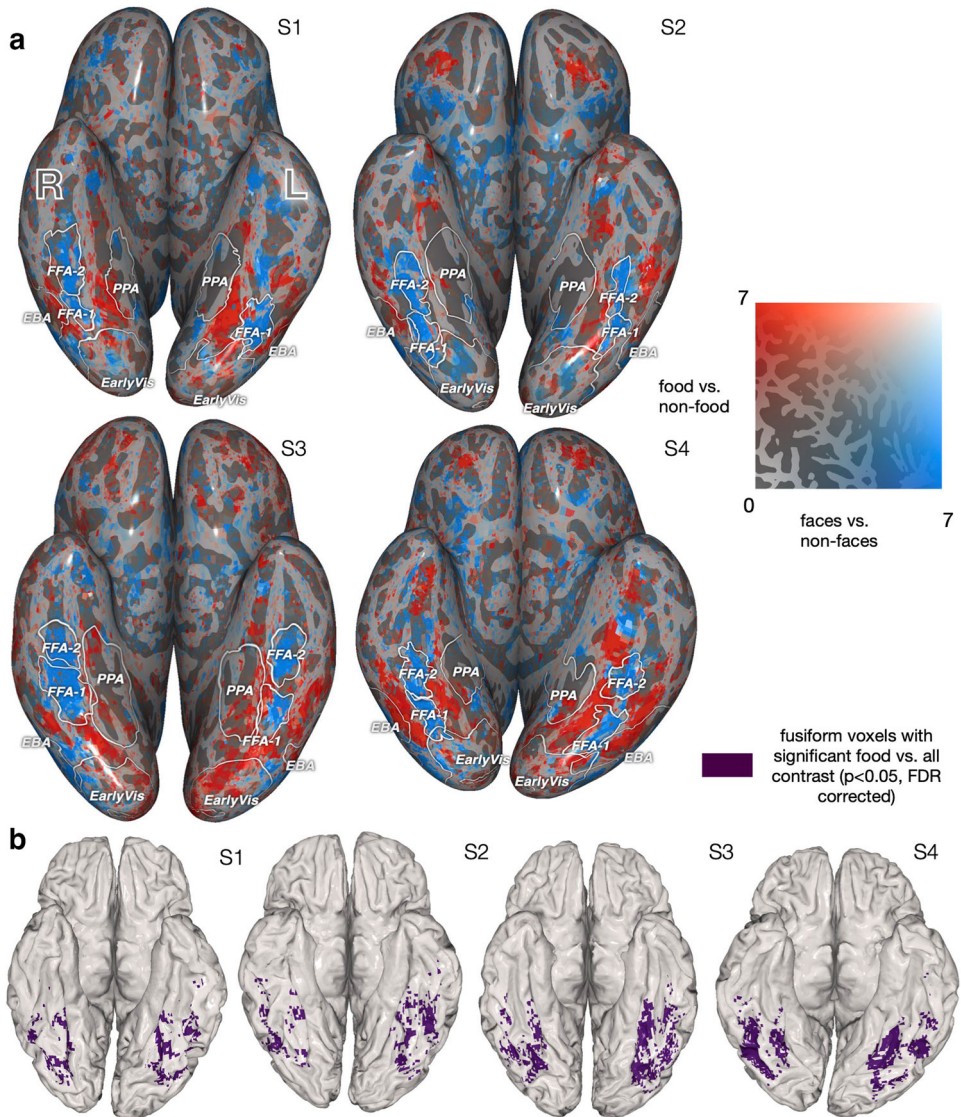

**Fig. 2 Experiment 1. Food-selective regions at the individual subject level. a** Comparing the spatial localization of food- and face-selective neural populations on the ventral surface, for S1–S4 (see Suppl. Fig. S1 for S5–S8). Voxels' *t*-statistics from two 1-sided *t*-tests comparing food vs. non-food (red) and face vs. non-face (blue). The regions identified by each contrast are largely non-overlapping. This pattern is maintained for food vs. non-(food and face) and face vs. non-(face and food) (Suppl. Fig. S5). **b** Spatial mask for food-selective regions used in subsequent analyses for S1–S4 (highlighting ventral visual responses). The mask is the overlap between the region that is identified from the *t*-test for food vs. non-food (**a**, red) at *p* < 0.05 (FDR corrected) and relevant neuroanatomically localized regions using the HCP atlas[23] (see *Methods*).

To understand the representational structure of these regions, we ran a principal components analysis (PCA) on the responses from all subjects to the shared food images. PCA was run using only voxels within the identified food region (Fig. 2) concatenated across all subjects (because the PCA was run on a concatenated dataset across subjects it does not account for the individual subject dependencies in the data; methods that handle multiple data tables, such as those presented in[26], might be used to test whether the group solution in our PCA is a good fit for all subjects), using responses associated with the shared food images only. This PCA produces for each voxel a set of principal component scores that capture the projection of its high-dimensional response profile across all images onto a lower-dimensional subspace. The axes of this subspace—shared semantic axes—corresponds to the dimensions in food image space that are most strongly reflected in the voxel responses (Fig. 4a). In Fig. 4b and c, we visualize the top and bottom images for each PC. The first three PCs are each associated with distinct

groups of voxels. PC1 is characterized by small positive patches around the center of each food-preferring strip on the ventral surface, with more negative values close to the edges of each strip. Negative and positive scores for PC2 differentiate the lateral and medial strips of the food-selective region. PC3 scores are generally more spatially diffuse, but in the right hemisphere, PC3 scores are more negative near the FFA (i.e., medial side of the lateral strip, lateral side of the medial strip). Based on inspection of the top and bottom images associated with each PC, PC1 captures the prominence of food in an image, distinguishing images with food as a key focus in the foreground vs. those with food as a background element. PC2 distinguishes food images based on overall scale, differentiating close-up images that focus on a few food objects from larger-scale images of food-related scenes (Fig. 4b). This is consistent with the pattern of positive scores for this PC on the medial side of the food-selective area, close to the PPA. PC3 distinguishes food images based on social attributes, separating food images that include few people from images of

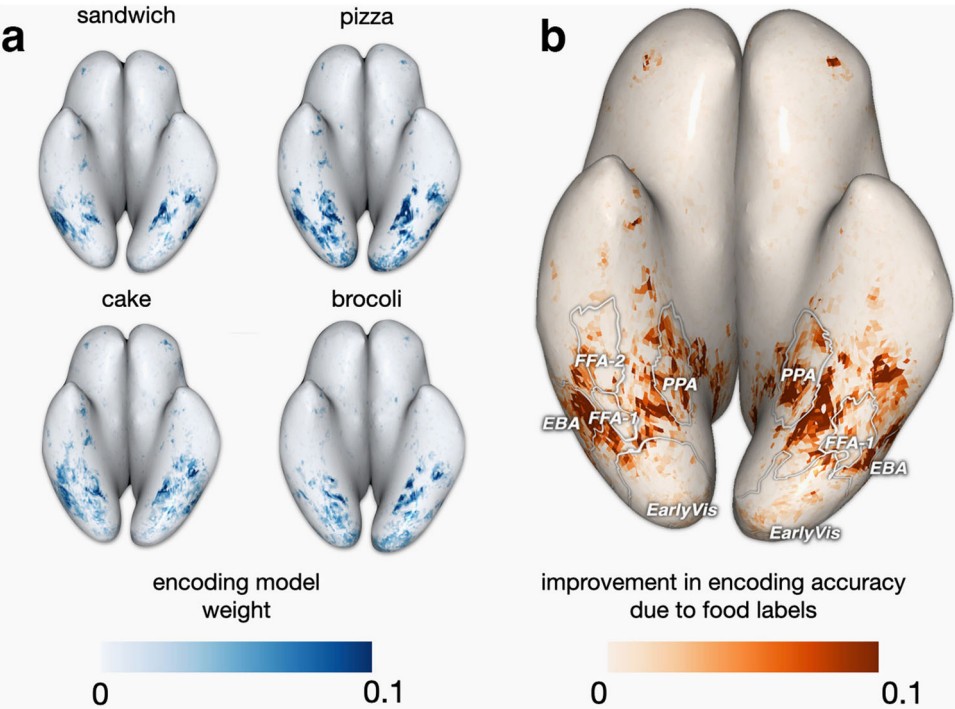

**Fig. 3 Experiment 1. A consistent set of food-selective regions can be identified across independent image sets with different labeling schemes.** We used the set of images for each subject that were not included in previous analyses, and an encoding model built from the 80 COCO object labels. **a** Voxel-wise encoding model weights for four food sub-categories from the original COCO dataset, shown for S1. We see variability in the weights, such as (perhaps, not surprisingly) pizza yielding higher weights in some areas than broccoli. **b** We compared predictive accuracy of an encoding model with all COCO labels (including 13 food and 67 non-food labels) to an encoding model with only the 67 non-food COCO labels. On S1's native surface, there is an improvement in validation set $R^2$ values when including the food labels ($R^2$ for the full model; $R^2$ for the model with food removed), with S1–S8 results in Suppl. Fig. S6. Weights corresponding to individual food labels (**a**) and the pattern of improvement in $R^2$ (**b**) highlight similar food-selective regions. Such consistent results lend further support for these regions being robustly food selective.

multiple people eating or preparing food, with social settings being at the end of the spectrum (Fig. 4c). Some amount of person or animacy-related information also appears to be reflected in the first two PCs (top right vs. bottom left images in Fig. 4b). Such results highlight the ecological importance of food as a category, as well as how high-level knowledge structures arise from the interaction between food and other ecologically important categories within the ventral visual cortex.

One concern with this interpretation is that the presence of people and/or faces in conjunction with food within our test images may lead to responses driven more by these well-known category-selective domains rather than food per se. That is, the PCs we obtain may be a result of the PCA picking up on food selectivity that overlaps with adjacent face- and body-selective regions. However, as we demonstrate in Experiment 2, we obtain the same pattern of food selectivity using "pure" food images. Similarly, Khosla et al. (2022) replicated their results (and our's) using BOLD5000[27]—a different dataset that includes primarily food alone images (images in BOLD5000 are drawn in part from ImageNet and include approximately 50 food categories). These results suggest that the presence of faces, people etc. is not a factor in the basic finding of food selectivity. Reinforcing this conclusion, as illustrated in Supplementary Fig. S7, when we reran the PCA excluding images that contained human faces and human bodies (but not the few animals that appeared with food), we observed essentially the same dimensional structure for PC1 and PC2 as seen for the full PCA, while PC3 became relatively less informative relative to the full PCA (possibly because the neural representation of food does incorporate a social dimension and this dimension was specifically excluded from this

secondary analysis due to the removal of images that depict social contexts).

Given these results, from a theoretical standpoint, we posit that the organization of food responses might reflect more than the adjacencies of food selectivity with other category-selective domains. Rather, the visual heterogeneity of food implicates non-visual factors as the underlying drivers in food-related visual responses. Based on prior work (e.g.[10,11]) and the adjacency of food-selective regions to the FFA and PPA, we propose that food responses are the aggregation of a wide variety of factors, including explicit social (face and body) and place associations, reward feedback circuits, and gustatory signals, all of which play some functional role in food processing as realized in the ventral visual cortex. Future work should explore this hypothesis, examining how the interaction of food with other systems gives rise to visual food selectivity.

To further explore what features drive the brain organization of food representations, we clustered food images according to their voxel responses in our food-selective regions. This analysis produces image clusters that are not easily characterized in terms of visual features, viewpoints or semantic attributes (Suppl. Fig. S8a). We also constructed image clusters using two neural-network models—CLIP[28] and ResNet-18[29]—from which we derived semantic and visual embeddings that did not include the associated brain activity for the images. CLIP is trained on both images and text captions, enabling us to extract features that capture the high-level semantics of the images. ResNet-18, trained solely on images and their associated object labels, yields features with less emphasis on scene semantics. As shown in Supplementary Fig. S8, the clusters arising from CLIP capture semantic

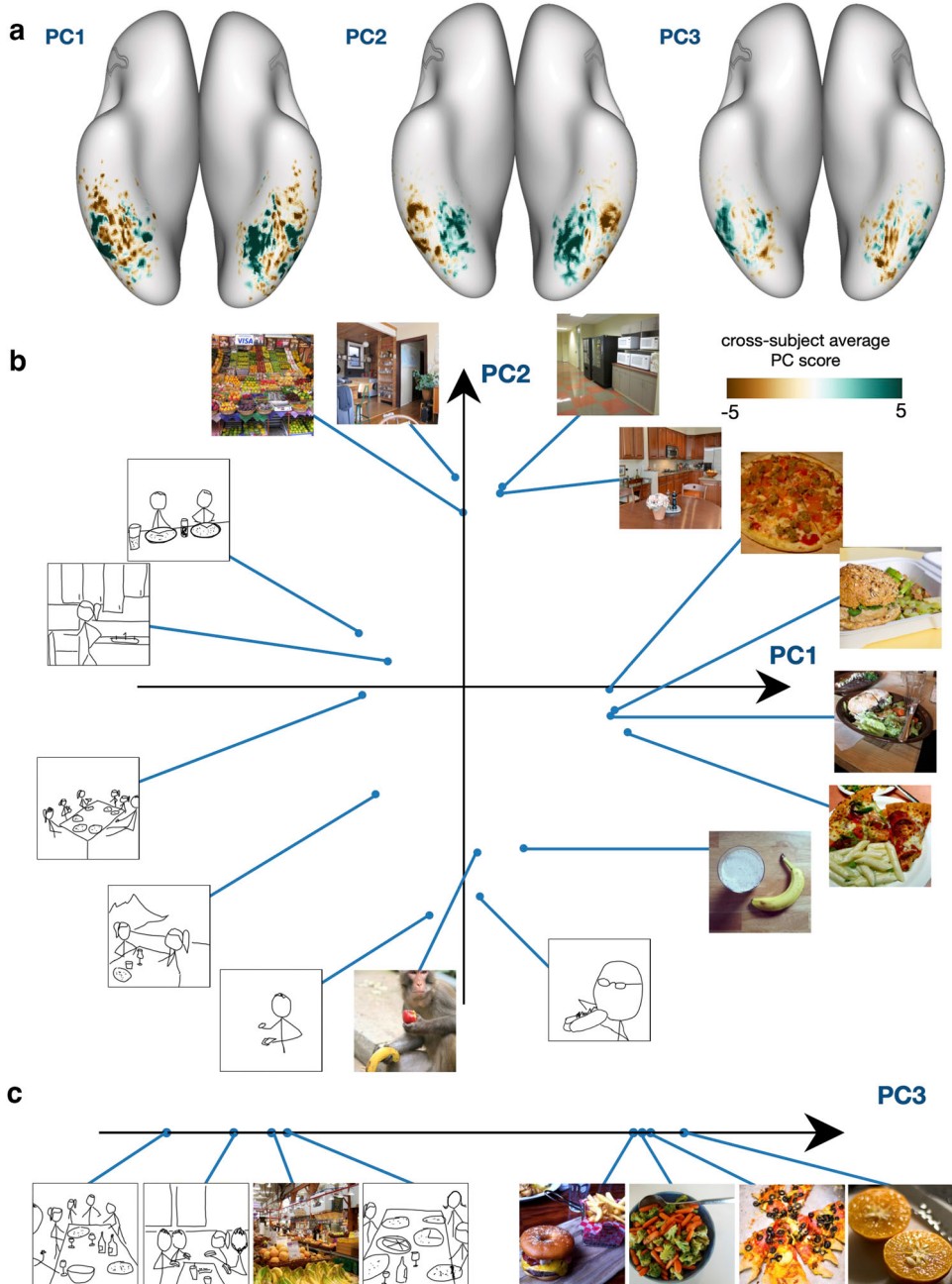

**Fig. 4 Experiment 1. PCA of responses from food-selective regions provides insight into their functional structure. a** Average principal component score across subjects for PC1, PC2, and PC3, shown on the MNI surface. Blue-green indicates high, brown indicates low PC scores. These top three PCs explain, respectively, 34.31%, 12.68%, and 11.16% of the variance. In (**b**) and (**c**), we show the images that lead to the highest and lowest activations in each PC. We include the 4 top and bottom images for ease of visualization. Top images for PC1 and PC2 are plotted in a 2D space (**b**), with the points connected to each image indicating its position in the space. In (**c**), we plot the top and bottom images for PC3 along a linear axis. Several patterns emerge here: PC1 scores yield small positive patches around the center of each food-preferring strip with more negative values close to the edges of each strip, and may capture the prominence of food in an image, separating images with focus on food in the foreground from those with food in the background. PC2 scores are higher medially (closer to PPA) and lower laterally, and seem to distinguish large-scale images of food-related places from close-by images of food and people eating food. PC3 scores in the right hemisphere food regions are lower at the center of the two strips, in the areas that border the FFA, while the left hemisphere does not show a clear pattern. PC3 appears to distinguish non-social food settings from social food settings. These results highlight that the combination of food with other ecologically important categories, including people (both faces and bodies) and places, creates a richer co-organization that reveals itself as gradients across cortex Due to licensing concerns, images showing people that were used in our study have been replaced with stick figures representing the structure of the original stimulus images. Replacement images (badly) drawn by MJT.

classes of food (e.g., fruits, deserts or meals; Suppl. Fig. S8b) while the clusters arising from ResNet-18 appear more visually organized and more focused on individual elements (e.g., broccoli, pizza; Suppl. Fig. S8c). Comparing the similarity of the cluster assignments of images for each of the three clustering procedures, neither CLIP or ResNet-18 clusters show any clear correspondence with our voxel-based clusters. The lack of correspondence in our clustering results suggests that the

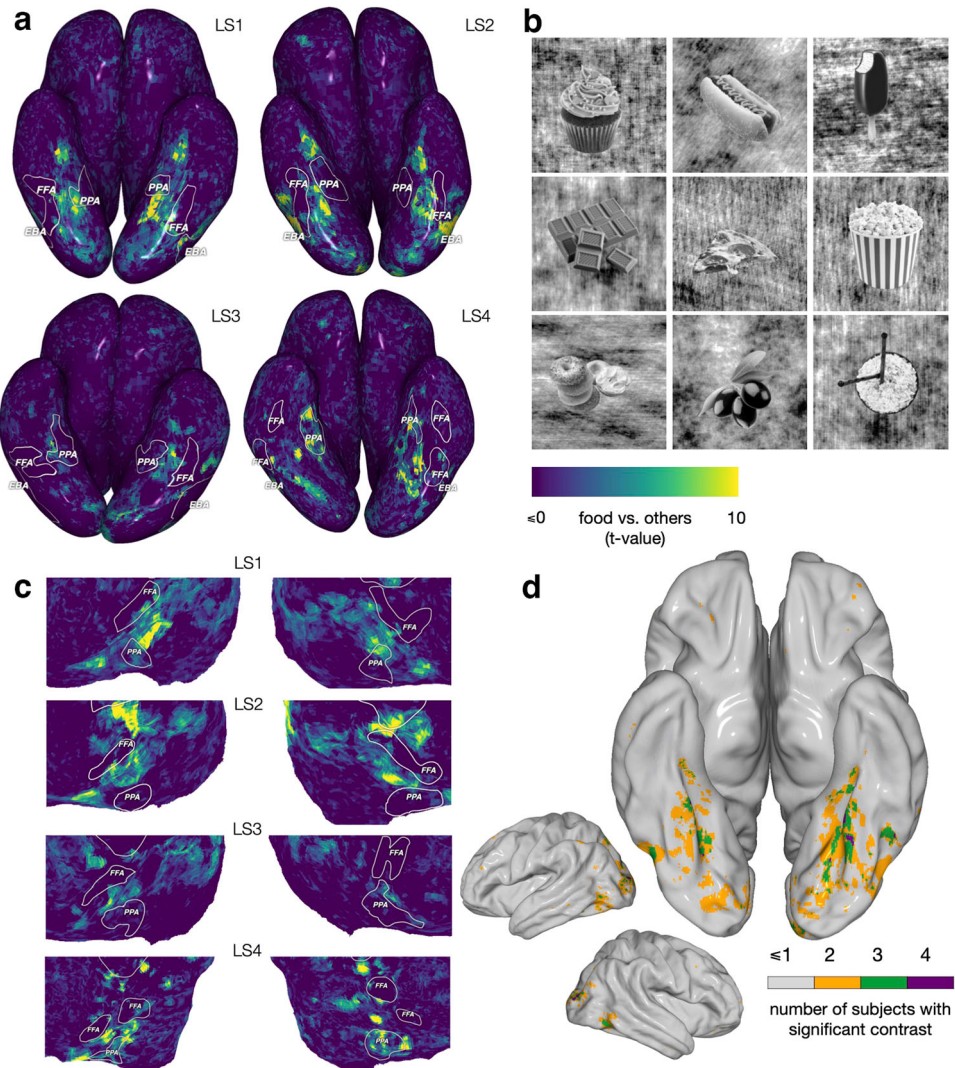

**Fig. 5 Experiment 2. Food-selective regions identified in an independent set of subjects using a visual localizer that includes grayscale images.** The fLoc localizer by Stigliani et al.[21] was adapted to include a food condition that was constructed by identifying images of food items from different categories and with different shapes, converting them to grayscale and superposing them on the scrambled images from the fLoc localizer (see *Methods*). Other conditions included faces, bodies, places and written words. **a** *t*-value of the food vs. other contrast shown on the cortical surface (viewed from the bottom) of each localizer subject (LS1–LS4). For each subject, the PPA, FFA and EBA were traced using the corresponding conditions in the localizer. Food-selective regions with a high value for the food vs. other contrast sit between the FFA and PPA of different subjects, with some subjects having high values on both sides of the FFA. See Supplementary Fig. S9 for the significance thresholds. **b** Examples of the stimulus images used in the food condition. **c** A cut-out of the flattened brain of each subject providing a different view of the food regions. There exists some spatial variability between subjects, but the relationship between the ROIs is more stable. **d** Semi-inflated lateral and semi-inflated bottom views of the MNI surface indicating voxels the subject count for a significant food vs. all contrast. Voxels more responsive to food are found in the dorsal visual cortex, with the highest concentration across subjects occurring in the fusiform visual cortex. This result replicates our initial finding with NSD (compare with Fig. 1c). As predicted, the location of the food region is spatially variable across subjects (see Suppl. Fig. S10 to compare with the variability of other classical localizers).

responses in food-selective areas do not organize easily into clusters based on visual similarity, scene semantics, or object semantics (and reinforces the complex, high-level nature of the dimensions found using PCA).

**Experiment 2: Hypothesis-driven analyses of food selectivity with controlled stimuli**. Our analyses using the NSD dataset allowed us to form a strong hypothesis on the presence of food-selective areas within the fusiform gyrus neighboring the FFA. Next, we designed a standard food "localizer" and collected new fMRI data in 4 new subjects to test whether we could replicate our results in a controlled experiment. We selected 82 images of different types of food with transparent backgrounds

from the https://www.stickpng.com/website. We converted the images to grayscale and superimposed them on images from the scrambled condition in the fLoc localizer[21] (Fig. 5b illustrates some examples). All images are shared in the supplementary materials folder. We included four additional conditions from the fLoc localizers: face (adults), body, place (houses) and words. We used the face vs. others, body vs. others and place vs. others contrasts to trace the FFA, EBA and PPA of each subject. In Fig. 5, we show the contrast of food vs. others for each subject on both their inflated and flattened surfaces. Supplementary Fig. S9 shows inflated and semi-inflated maps illustrating the voxels for which the contrast is significant for each subject ($p < 0.001$, FDR corrected). As in Experiment 1, we considered the magnitudes of these responses as realized in a measure of voxel-wise selectivity.

As illustrated in Supplementary Fig. S12a, consistent with the faces vs. other contrasts shown in Fig. 5, the pattern for each functional region is aligned with both the results from Experiment 1 and with prior results establishing selectivity for faces and places. Equally important, as illustrated in Supplementary Fig. S12b, both the magnitude and distribution pattern of voxel-wise selectivity for food is on par with that for faces and places.

Instead of computing a group analysis that would smooth the results in space, we treat every one of the 4 subjects as a replication unit for the hypothesis of food selectivity[30]. The results of the localizer replicate the findings of Experiment 1 (and[19,20]) in every one of the four subjects: statistically significant food-selective regions fall within the fusiform gyrus adjacent to the FFA. Our new results also indicate that factors such as color or food appearing in a natural scene are not essential for obtaining selective activation for food (in parallel work, Khosla et. al.[20] likewise predicted that food-selective responses should be obtained using grayscale images; see their Fig. 4c). While some spatial variability in food-selectivity exists across subjects, regions with high values for the food vs. other contrast lie between the FFA and PPA of different subjects, with some subjects having high value voxels on both sides of the FFA. After converting the subjects' results to MNI space and counting the number of significant voxels in each MNI location, we see less spatial agreement among subjects in the food vs. other contrast as compared to the face, body, and place contrasts (face vs. all, body vs. all, place vs. all, and words vs. all; see Suppl. Fig. S10; in parallel work[20], observe a similar pattern as illustrated in Suppl. Fig. S5 of their paper). More specifically, for each of these other contrasts there exists a region in which all subjects show a significant effect. However, for the food contrast, we find greater spatial variability: at most 3 of 4 subjects have a significant contrast in the same region of the left fusiform, and only a small number of voxels show a significant contrast across all subjects. This result is aligned with our findings using NSD in Experiment 1, where the most consistent region is one in which only 5–6 of the 8 subjects showed a significant contrast (Fig. 1). Such spatial variability may be one important reason why earlier studies—particularly those relying on group analyses—may have failed to identify regions selective for food.

## Discussion

How are knowledge representations organized in the human brain? Within the visual system, one of the hallmarks of the past several decades has been *category selectivity* for faces, bodies, places, and words[1–5]. Consistent with the ecological importance of these categories, we identified selectivity for another ecologically relevant category, food, within the ventral visual stream. In our present study, we used both data- and hypothesis-driven fMRI methods. Two parallel studies[19,20] also used data-driven methods applied to the same large-scale natural scenes dataset[17] and confirmed our finding of food selectivity in Experiment 1. Our study also provides a range of analyses not included in these other studies, as well as new and informative data from a second, hypothesis-driven experiment. First, using NSD, we show that the identified food regions are maximally activated by food images. Second, we establish that food-selective responses do not appear to be confounded with image viewpoint (zoom, reach or large-scale). Third, we find that the inclusion of food-related category features in an encoding model leads to improved prediction accuracy in food-selective regions. Fourth, we demonstrate that PCA can be used to uncover both the large-scale topography within food-selective areas and the interaction of food coding with other semantic dimensions. Namely, we find that the representation of food in the food regions appears to be organized

in gradients across cortex that relate food to other important information processed nearby (social and place-related information). Fifth, we verify the robustness of food-selectivity by showing consistent food-selective responses across independent sets of NSD images, and we provide the first characterizations of the fine-grained structure of representations within the food category itself. Finally and uniquely, we directly validate these results using hypothesis-driven methods in the form of a standard "localizer" that included grayscale images of food. The results of Experiment 2 replicate our results with NSD and provide direct evidence that color is not a confound in food-selectivity. Equally important, these results also suggest that food-responsive regions are more spatially variable across individuals than other visual functional ROIs, thereby helping us reconcile current findings of food selectivity with previous failures and with claims of overlap between food- and face-selective regions.

Although our focus was on selectivity in the ventral visual system, we note that we also observed food selectivity in the parietal and frontal cortices in Experiment 1; however, the localization of these regions was less consistent over subjects (Fig. 1c) and did not replicate when we used our context-free localizer images (Fig. 5). Other brain regions may also play a role in processing food information, particularly during visually-guided behavior. The dorsal visual areas in particular may process the actions or affordances associated with food (i.e., cooking/eating), as suggested by past work showing that object representations in dorsal visual cortex tend to be action-oriented[31,32]. Activation in frontal cortex appears to overlap roughly with orbitofrontal regions (semi-inflated bottom view map in Fig. 1c), which may reflect the involvement of these areas in processing reward information associated with certain foods[13,16,33,34]. Food selectivity was also observed in a number of subjects in the insular cortex, which has previously been implicated in taste processing[13,34]. While our paper focuses on visual selectivity for food in the fusiform cortex, future work should investigate the interaction of the visual food selective area with these other areas, perhaps using manipulations that vary reward or action representations evoked by food stimuli.

Our approach and results allow us to rule out several alternative explanations for the finding of food selectivity. It is not likely that food selectivity reflects preferential responses to "reachspaces"[7], rather than food per se. This is ruled out on the basis that our labeling taxonomy allowed us to control for image perspective (i.e., including *reach* as a label). Specifically, we found that food-selectivity remained stable even after removing the *reach* labeled images. Another possible alternative is that food-selectivity reflects preferential responses to small vs. big real-world object size[6], again, rather than food per se. However, the representation of real-world object size manifests as *big* flanking the medial side of the FFA and *small* flanking the lateral side of the FFA. Thus, this explanation can be ruled out in that our observed food selective responses co-locate more with big, as opposed to small, regions, yet food categories, particularly prepared foods, have small real-world size. A third possibility we can reject is that food selectivity can be solely attributed to greater attention or higher intrinsic visual salience for food relative to non-food[35]. Both human faces and bodies are subject to the same kinds of saliency effects[36], yet attentional/saliency differences are not the preferred explanation for face or body selectivity[37]. Moreover, within our study, faces and bodies comprised a reasonable proportion of the non-food contrast images, yet food selectivity was robust across these comparison categories (as is also the case in ref. [16]). Finally, it is not likely that low- or mid-level visual features (i.e., spatial frequency, curvature, texture) underlie our pattern of results. This is supported by the fact that food selectivity was primarily found in higher visual areas, rather

than early visual areas (Fig. 2). The visual variability of food makes it unlikely that there is a set of low- or mid-level visual features or high-level shape structures that consistently correspond to food (in contrast, see ref. [38–40]). Finally, as discussed below, another explanation that we can reject is that our food selective responses are mostly driven by color.

These conclusions are also consistent with a recent MEG study which excluded low-level visual features as an explanation for food selectivity[41]. Similarly, the two recent papers that likewise identified food selectivity using the same dataset we used here included several analyses that help to rule out a variety of low- or mid-level features as the basis for the observed selectivity[19,20]. Of note, both papers reported an intriguing overlap between food selectivity and color-biased brain regions. While Pennock et al.[19] favor an account in which color is a feature common to food and, as such, food-selective regions may respond to color even in the absence of food inputs, Khosla et al.[20] explicitly include color in their analyses and conclude that food selectivity cannot be explained by color alone. They do, however, acknowledge that selectivity for food and color-biased responses are "linked". As in Pennock et al.[19], they suggest that color is important for the identification[42], evaluation[43], and selection of food[20]. Our new data sheds conclusive light on this question. While color may be an important part of learning new food categories, the results of Experiment 2 demonstrate that food-selective responses can arise in the absence of color (Fig. 5). What remains to be determined in future work is whether the functional role of color in food-related behaviors leads to the instantiation of color biases in food-selective regions or whether color biases are present absent food selectivity and, as such, may help facilitate the acquisition of food representations in these regions.

Past work has presented conflicting accounts of the degree of overlap between food-selective and other category-selective visual regions[16]. Claims of overlap are questionable in that they were based on group-level analyses and any overlap may have been an artifact of the variability in the localization of food-selective regions within individuals (which may arise in part from the high visual variability of food as a category). In particular, Adamson and Troiani[16] claimed that "there is overlap in face and food activation within the fusiform and that this is spatially consistent at the group level." This inference is puzzling in light of the fact that the same study presents a visualization of peak coordinates for both face and food clusters in individual subjects that appears to show separation between the two regions of selectivity (Fig. 2 of[16]). However, Adamson and Troiani focus on across-subject tests in order to support the claim that food selectivity co-localizes with face selectivity. This leads them to conclude, we believe incorrectly, that food and face recognition share a common neural substrate and, presumably, common underlying computational mechanisms.

The difference in our Experiment 1 results vs. those of Adamson and Troiani[16] may be due to our use of a more sensitive within-subject, voxel-wise analyses. Across multiple methods and within 8 individual subjects, our results indicate that food and face selectivity do not co-locate (Fig. 2a, Suppl. Fig. S1 and Suppl. Fig. S5). Reinforcing this separation between regions, in our Experiment 2, there is almost no overlap between food- and face-selective areas in individual subjects (Fig. 5a and c). In this same experiment, the cluster of voxels with the greatest consistency across subjects is in the left fusiform directly adjacent to the FFA (Fig. 5d). We note that group averaging—as used in[16]—could potentially blur these significant food- and face-selective areas so as to create the appearance of overlap at the group level within left fusiform (as reported in ref. [16]). Consistent with this interpretation and the results of our Experiment 2, Adamson and Troiani[16] report separation in the peak coordinates for face and

food clusters for individual subjects. Similarly, Khosla et al.[20] found that the food-selective component of their results was less spatially correlated across subjects as compared to other category-selective components. Consequently, there is little evidence to support a claim that food and face representations arise from the same fine-grained principles of visual processing. Rather, there is variability in the localization of food-selective regions across subjects; as such, it is critical to assess selectivity on an individual basis.

More generally, why have most previous efforts to localize a food-selective region of ventral cortex failed (e.g., P. Downing and N. Kanwisher, 1999, Cogn. Neurosci. Soc., poster; based also on multiple anecdotal reports of similar failures)? A variety of factors may have impacted the results (or lack of results) in many of these prior studies. One possibility is that some of the apparent inconsistency in detecting food-selective responses is, in part, due to relying on isolated, somewhat unrealistic food and non-food images (e.g., Downing et al.[14]). However, Experiment 2 identifies food selectivity using grayscale images of food. As such, while the naturalness of the COCO images used in NSD may enhance food-selective responses, it seems unlikely that naturalness alone (nor the absence of color) can account for prior failures. At the same time, it is worth noting that both Adamson and Troiani[16] and Tsourides et al.[41] used naturalistic food images and were able to successfully identify food-related neural responses as measured by functional MRI (fMRI) and magnetoencephalography (MEG), respectively.

A second factor contributing to earlier null results may be that prior studies used an insufficient number of food images, thereby failing to capture the large variety of visual properties of food or of the natural contexts in which food appears. Unlike faces, bodies, or word stimuli, food images vary widely in low- to mid-level visual characteristics such as curvature, shape, texture, color or the organization of the parts into a whole. Thus, greater numbers of food stimuli not only increase experimental power in and of itself, but lead to better coverage of "food appearance space" as it may be mentally and neurally represented.

A third factor which may have made identifying food-selective regions more challenging is potential variability across individuals in the neural localization of food-related responses—a prediction supported by the individual variability seen in the results of Experiment 2 (and in[16] and[20]). One possible reason for this variability may be that voxels processing food are interleaved with voxels processing other object related properties[20]. Given such variability, group-average analyses (Fig. 5d) are unlikely to reveal a consistent, significant cluster of separable food-selective voxels across subjects. However, when one considers individual subject responses, not only do significant food-selective voxel clusters emerge, but we observe *spatially relative* consistency between these clusters and other functionally-localized ROIs (Fig. 5c). For this purpose, we make our food localizer available for the community, along with the code to process it and generate visualization using the pycortex software[44].

Another possible reason for individual variability in food-selective regions is the visual heterogeneity of food. Indeed, this latter point has been raised as one reason why a food-selective visual region seemed unlikely—in contrast to visually-homogeneous categories such as faces and written words, foods vary dramatically in shape, texture, and color. Consequently, it is unclear how a single visual mechanism might learn across this appearance diversity. One potential solution has been articulated in modern machine learning where building a classifier for a complex class comprised of multiple sub-classes with an inherent organization (such as food) is a problem referred to as hierarchical classification[45,46]. One of the common ways hierarchical classification is solved is by combining the predictions of

specialized classifiers for each of the different sub-classes into a single prediction[47]. Thus, one can conceive of food-selective responses as a set of specialized classifiers for different food sub-types. Given the complexity of such representations, as well as their potential interactions with culture, taste[16], and experience[11], spatial variability in food-selective responses is not surprising. Based on this logic we are exploring whether other visually heterogeneous categories with high reward or social significance may, similar to food, come to be selectively represented—possibly intermixed with food representations—in ventral visual cortex.

Fourth, as just discussed, the visual heterogeneity of food may lead to food-selective regions that are more spatially distributed as compared to other category-selectivity responses, possibly including multiple sub-regions. To boost statistical power, standard neuroimaging analyses often forgo individual-level statistical tests in favor of across-subject tests that are biased to "blur" localized responses[16,48]. These analyses are more likely to identify regions that are well aligned across subjects[49]. In contrast, in both of our experiments, we rely on within-subject analyses that are better able to pull out category-specific neural responses for individual brains (reinforcing this point, see the demixing analyses in[20]). Moreover, in Experiment 1, our fine-grained analyses reveal that the top images in food regions overwhelmingly depict food. Thus, while it is possible that the neural representations of other categories are intermingled with the representation of food, our results favor distinct, but perhaps distributed, food-selective regions within ventral visual cortex. However, at a finer grain, it also possible that there is some spatial overlap between food-selective neural populations and neural populations selective for other categories, thereby "diluting" the food-drive responses of individual voxels (see Fig. 6[20]). As a coda to our use of more sensitive data analysis tools, we note that modern fMRI measurements are much improved over earlier experiments. For example, Experiment 1 used NSD which was collected using a 7T scanner and high resolution temporal and spatial sampling[17,50], while Experiment 2 used a state-of-the-art 3T scanner and 64 channel head coil.

Finally, while a finding of food selectivity naturally emerges from considering ecologically important visual categories, this leaves open the question as to how such selectivity arises in the human brain. We speculate that, similar to human language, domain-relevant perceptual inputs related to food can vary widely depending on the cultural and physical environment. Learned representations for food are only loosely constrained at the surface level, but still reflect common underlying mechanisms that have emerged over the course of evolution due to reward and the selection for learning abilities that flexibly responded to variations in inputs (the "Baldwin Effect"[51,52]). Thus, as a core property of knowledge organization, food selectivity is likely to have emerged as a neural preference shaped heavily by semantic associations, context, and reward.

## Methods
### Experiment 1
*fMRI data.* We used the Natural Scenes Dataset (NSD)[17], consisting of high-resolution fMRI responses to natural scenes. The detailed experimental procedure are described by Allen et al.[17]. The naturalistic scene images were pulled from the annotated Microsoft Common Objects in Context (COCO) dataset[18]. 8 subjects each viewed between 9000 and 10,000 natural scene images over the course of a year, each repeated 3 times. Of the 70,566 total images presented, 1000 were intended to be viewed by all subjects. However, because some subjects dropped early, they did not all view the 1000 images 3 times. For the purposes of this paper, we use any of the 1000 images for a subject if it was viewed at least once (515 were seen three times by each subject, 766 were seen at least two times and 907 at least one time). Thus, for subjects S1–S8 we use respectively 1000, 1000, 930, 907, 1000, 930, 1000, and 907 shared images.

The data were collected during 30–40 scan sessions. Images were square cropped, presented at a size of 8.4° × 8.4° and for 3 s with 1s gaps in between images. The subjects were instructed to fixate on a central point and to press a button after each image if they had seen it previously.

The functional MRI data were acquired at 7T using whole-brain gradient-echo EPI at 1.8 mm resolution and 1.6s repetition time. The preprocessing steps included a temporal interpolation (correcting for slice time differences) and a spatial interpolation (correcting for head motion). Single-trial beta weights were estimated with a general linear model. FreeSurfer[53,54] was used to generate cortical surface reconstructions to which the beta weights were mapped. The beta weights corresponding to each image were averaged across repetitions of the image, resulting in one averaged fMRI response to each image per voxel, in each subject.

The dataset also included several visual ROIs that were identified using separate functional localization experiments. We drew the boundaries of those ROIs for each subject on their native surface for better visualization and interpretation of the results. All brain visualization were produced using the Pycortex software[44]. We create flattened, inflated and semi-inflated maps by setting the 'unfold' parameter to 1, 0 and 0.25 respectively. Fig. 1 and Fig. 2 show the left and right hemisphere for each type of view we show (flatmaps and semi-inflated or inflated bottom and lateral views). These conventions are maintained across all brain plots in the manuscript and supplemental materials.

*Image labeling.* The authors and a graduate student in our labs ($n = 8$) performed manual image labeling for the 1000 potentially shared images based on each image's depicted location, image perspective and content. Location refers to whether the image is indoor or outdoor (or ambiguous), content refers to the categories of objects in the image (including the binary existence of food), and image perspective refers to the approximate scale of the image, discretized into *zoom*, *reach* or *large-scale*. *Zoom* refers to a very close shot, thereby likely concentrated on one object and excluding surrounding information. *Reach* images display objects at a human-reachable distance, and may activate representations related to object affordances[7,22]. *Large-scale* images encompass the remaining images, which include an image of a typical scene as opposed to one or more close-up objects. Images could only be assigned one label for location and perspective, but could be assigned multiple content labels. More details about this image labeling are described in the Fig. 1a and b and Supplementary Table S1. Labeling was performed using the Computer Vision Annotation Tool[55]. In order to avoid variation in labels and ensure consistency, we performed several rounds of labeling and verification across multiple raters; each image was seen by a least two raters. Disagreements were discussed in the group of raters until unified labeling assignments were reached.

*Encoding models.* We constructed two different encoding models. The first was based on our hand-labeled annotations of the 1000 potentially shared images (Fig. 2). Encoding all 16 hand-labels into a single binary vector per image, we utilized voxel-wise ordinary least squares (OLS) encoding models to predict each individual voxel response to a given stimulus. Identifying voxels more responsive to category $A$ over other category was done using a 1-sided $t$-test between the respective learned model coefficients for category A vs. the coefficients for the other categories, as is done in a typical generalized linear model (GLM) analysis. Note that this analysis collapses across the three "attributes" used in our labeling taxonomy (i.e., food is compared against object categories like faces, as well as against location labels like indoor). We used these methods to identify voxels that are more responsive to food than other labels, as well as for face vs. other labels. We obtained a $p$ value from the $t$-value, then corrected for multiple comparisons across all voxels using the Benjamini-Hochberg False Discovery Rate procedure (FDR)[56], which is appropriate for fMRI results due to the assumption that they possess positive dependence[57,58]. The significance of the contrast was computed at the subject level, the results were converted to MNI space, and the sum across subjects was plotted in Fig. 1c. Pycortex was used for transformation to MNI space of each subject's result. It relies on the Flirt tool[59–61] from FSL[62].

Our second encoding model was based on COCO object category labels, and made use of the set of images that were unique to each subject (Fig. 3). The purpose of this model was to verify that our proposed food region derived from the shared images is consistent across the larger set of images that also includes images not used in the first analysis. We used the 80 COCO object category annotations provided in the dataset, specifically each COCO label's corresponding bounding box proportion relative to the image (i.e., proportion of the image covered by the category of interest), as input to a ridge regression encoding model. We built and tested the model via 10-fold cross-validation, where $R^2$ was computed on a tenth of the data not used for training at each fold, and the 10 resulting $R^2$ values were averaged. The penalty parameter for each voxel was chosen independently by nested $10 - $ fold cross-validation. When determining which images were used to fit the encoding model, we create a set of images that contained half food and half non-food images. We considered images to include food if their maximum food label proportion exceeded a threshold of 0.15. We identified 940 such images, and randomly selected 940 non-food images, together creating a total input set of 1880 images. We built two models, one with all the labels, and one with all the labels that were not food (67 in total). We then computed the voxel-wise $R^2$ improvement from including food labels in the regression. In addition to helping identify voxels that responded most to inclusion of food, this encoding model also helped us visualize food sub-category activations. We observed the voxel-wise learned weights corresponding to specific COCO food labels (i.e. cake, sandwich) to uncover potential food sub-category patterns.

*Decoding models.* While an encoding model is able to provide some insight into single-voxel selectivity through response predictions, a decoding model can

uncover distributed pattern-level representations of visual features. To observe representations at the population level, we used a searchlight decoding method[24]. Specifically, for each voxel in the cortical sheet, we defined a searchlight sphere that consisted of 27 nearby voxels, and we trained a decoder to classify the existence of food based on the pattern of activation across these voxels. We used fivefold cross validation via Support Vector Classification, with our input image set consisting of 108 food images and 108 randomly selected non-food images from the shared images. High decoding accuracy from this method suggests that an area encodes food-related information at the pattern level, which our model is able to exploit in order to classify the existence of food.

*Determining the ventral visual food selective regions.* To generate a mask that only included the ventral visual food selective region, we first manually selected apparent relevant ROIs via the Glasser HCP Atlas[23]. We use the concatenation of sub-areas TE2p, PH, VVC, v8, PIT, FFC, and VMV3 to create our mask. After converting the mask for this anatomical area into each subject's native space, we identified the intersection of this mask with the identified food region from a food vs non-food significance test (Fig. 2 shows the final mask definition).

*Principal component analysis (PCA).* We ran PCA to better understand possible structure and/or correspondence in these food-selective regions. Using the food mask above that consists only of our proposed food region, we selected 'food-relevant' voxels for each subject. Then, we ran PCA on a matrix of concatenated 'food-relevant' voxels for all subjects (rows) by the activity related to shared food images (columns), reducing along the image dimension (the columns). That is, the number of rows consists of all previously identified food selective voxels from all subjects concatenated together (i.e., if we have 200 food voxels for each subject, then the matrix across 4 subjects will have 800 rows). Thus, the matrix on which we ran PCA is two dimensional (# of voxels by # of images) and we have only one PCA model for all subjects. We extracted the top principal axes of this matrix, and projected our initial data matrix onto the calculated lower-dimensional space to obtain the voxel-wise PC scores on the brain. To compare the voxel-wise PC scores across subjects, we converted the scores for each subject to the MNI template and average the scores across subjects for each MNI voxel. We identified the most positive and negative contributing images to each axis by computing the dot-product between the PC score and the activity related to an image, to assess whether the representations of each principal axis were cohesive or semantically interpretable.

*Clustering analyses.* We ran a K-means clustering analysis to better investigate visual and semantic patterns in the food selective regions. As a point of comparison with the voxel clustering results, we also clustered visual and semantic embeddings of these images derived from deep neural networks. To compute the clusters for one subject, we picked 940 food images. Voxel embeddings were calculated for each individual subject, using responses from voxels within the ventral food mask. To obtain visual and semantic embeddings for these same 940 images we used two trained deep neural networks: CLIP and ResNet-18[28,29]. CLIP, trained on both images and text, allows us to extract features arising from a contrastive learning paradigm with dual semantic and visual constraints. We used the pretrained ViTB32 model, which was trained to align image and text embeddings within a shared space. Within this model, we extracted the features given an input image from the vision module of the model. Given an image, we call these corresponding CLIP features the CLIP embedding.

ResNet-18, trained on solely images, provides a visual feature-based embedding with no language component. Given an image, we ran a ResNet-18 model pretrained on ImageNet to extract the features from the average pool layer immediately preceding the final fully-connected layer[63]. We refer to these extracted features for a given image as the corresponding ResNet embedding of that image.

To cluster embeddings, we used K-means clustering algorithm with Euclidian distance. We consider a range of $K$ values and for each, observe the average Euclidian distance from each data point to their corresponding cluster centroid. Next, we selected the first $K$ value that led to the drop in the average distance for voxel embeddings beyond which the decrease plateaus (the elbow method). This value was 4. We use this same $K = 4$ for all three embedding clusterings.

To compare different clustering assignments, we constructed for each clustering procedure a $940 \times 940$ matrix where the rows and columns correspond to the 940 images. Each cell in this matrix is an indicator value where $matrix_{i,j}$ is 1 if the two images $i$ and $j$ are in the same cluster, and 0 otherwise. We then used Pearson correlation to compute the correlations between two clustering assignments. To visualize each cluster, we chose the closest images to the centroid of that cluster.

### Experiment 2

*MRI data collection.* MRI data were acquired on a 3T Siemens Prisma MR scanner at the BRIDGE center at the Carnegie Mellon University campus using a 64-channel phased array head coil.

*Functional Images.* Functional images were collected using a T2*-weighted gradient recalled echoplanar imaging multi-band pulse sequence (cmrr_mbep2d_bold) from the University of Minnesota Center for Magnetic Resonance Research (CMRR)[64,65]. Parameters: 68 oblique axial slices co-planar with the AC/PC; in-plane resolution = 2 × 2 mm; 106 × 106 matrix size; 2 mm slice thickness, no gap; interleaved acquisition; field of view = 212 mm; phase partial

Fourier scheme of 6/8; TR = 1500 ms; TE = 30 ms; flip angle = 79 degrees; bandwidth = 1814 Hz/Px; echo spacing = 0.68ms; excite pulse duration = 8200 microseconds; multi-band factor = 4; phase encoding direction = A to P; fat saturation on; advanced shim mode on. During functional scans, eyetracking was acquired using an EyeLink eye tracker.

*Anatomical Images.* A T1 weighted MPRAGE scan was collected for each participant. MPRAGE parameters: 208 sagittal slices; 1mm isovoxel resolution; field of view = 256 mm; TR = 2300 ms; TE = 2.03ms; TI = 900 ms: flip angle = 9 degrees; GRAPPA acceleration factor = 2; bandwidth = 240Hz/Px.

*Subjects.* Functional data were collected from four subjects (3 female/1 male) aged 21–29. All subjects were healthy and had corrected to normal vision. Written informed consent was obtained from all subjects and the study was approved by the Carnegie Mellon University Institutional Review Board.

*Paradigm.* The data were collected in runs of length 4 min. Subject LS1 underwent 4 runs of the localizers, while subjects LS2-LS4 underwent 9 runs. We did not see a difference that appeared to be driven by the amount of data.

We selected 82 images of different types of food with transparent backgrounds from the https://www.stickpng.com/website. We converted the images to grayscale and superimposed them on images from the scrambled condition in the fLoc localizer[21] (Fig. 5b illustrates some examples). All images are shared in the folder referred to in the *Data Availability Statement* section. We used the mini-block design (duration = 6s) proposed by Stigliani et al.[21]. Along with the food condition, we also use the adult condition (to define faces), the house condition (to define places), the word condition and the body condition. We use the first 82 images from each condition provided in the localizer to have the same number as the food images. We adapted the Stigliani et al.[21] code to present our stimuli. The code uses Psychtoolbox-3[66–68] and runs on Matlab.

Images were square, presented on a gray background at a size of 11.4° × 11.4° visual angle on a BOLDscreen32 LCD Display and for 0.5 s each. The subjects were instructed to fixate on a central point and to press a button if they see a repeated image (1-back task).

*Data preprocessing.* Each subject's native surface was reconstructed using Freesurfer[69]. Functional scans were motion corrected using SPM12[70]. Through Pycortex, alignment of the functional data to the structural data was obtained (using bbregister from Freesurfer). Our code pipeline (provided in the folder referred to in the *Code Availability Statement* section) includes detrending and lightly smoothed with a Gaussian kernel of standard deviation 1 mm, using standard functions part of the scipy package[71]. Pycortex was used to mask the cortical data (by relying on maps estimated by Freesurfer). Pycortex was also used for transformation to MNI space. It relies on the Flirt tool[59–61] from FSL[62].

*Encoding models.* We followed the same procedure used with the shared NSD images to compute a contrast between condition A and other conditions after estimating voxel-wise ordinary least squares (OLS) encoding models. We computed a t-value for each of the "food vs. other", "face vs. other", "body vs. other", "place vs. other" and "word vs. other" contrasts. We used the Benjamini-Hochberg False Discovery Rate procedure (FDR)[56] and $\alpha = 0.05$ to identify significant voxels for each contrast of each subject at each voxel.

We drew the boundaries of the FFA, EBA and PPA for each subject using the "face vs. other", "body vs. other" and "place vs. other" significance maps, respectively. This enables us to better understand the "food vs. other" contrast results. Note that unlike in NSD where the ROIs labeled using separate data, here the data from the "food vs. other" contrast is the same as the one used to draw the other ROIs. The significance of the contrast was computed at the subject level, the results were converted to MNI space, and the sum across subjects was plotted in Fig. 5. The same was repeated for the other contrasts which can be seen in Supplementary Figure S10.

*Statistics and reproducibility.* Statistical analyses were performed using Python and data visualizations were accomplished using Pycortex[44]. Significant voxels for contrasts within encoding models were identified by computing a 1-sided t-value for each contrast. We then obtained a $p$ value from each $t$ and corrected for multiple comparisons using the the Benjamini-Hochberg False Discovery Rate procedure (FDR)[56] and $\alpha = 0.05$. Voxel-wise selectivity for each visual category was computed as described in the captions of Supplementary Figs. S11 and S12. In Experiment 1, 8 subjects viewed between 9000 and 10,000 natural scene images (with between 907 and 1000 being shared across all subjects)[17]. In Experiment 2, 4 subjects viewed 360 stimulus images (comprised of food, faces, houses, words, and bodies) per run (either 4 or 9 runs). Additional information on the analyses are provided in *Methods*.

**Reporting summary**. Further information on research design is available in the Nature Portfolio Reporting Summary linked to this article.

## Data availability

The NSD data was made available by Allen et al.[17]. The localizer data are available at https://kilthub.cmu.edu/articles/dataset/Selectivity_for_food_in_human_

ventral_visual_cortex/22049177 and the code to process them is part of a public Github repository at https://github.com/brainML/food4thought. All other data are available from the corresponding author on reasonable request.

## Code availability

Our code is available as a public Github repository https://github.com/brainML/food4thought.

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

## Acknowledgements

Collection of the NSD dataset was supported by NSF IIS-1822683 and NSF IIS-1822929. The authors thank the NSD team for collecting and sharing the dataset. MMH was funded by a Distinguished Postdoctoral Fellowship from the Carnegie Mellon Neuroscience Institute. JP was supported by NSF BCS-1658278. The authors thank the following people for contributing ideas and commentary to this project: Laurie M. Heller, Andrew Luo, Isabel Gauthier, Marlene Behrmann, and Brad Mahon. The authors also thank John Pyles for help with setting up the localizer.

## Author contributions

N.J., A.W., M.M.H., R.L., J.S.P., M.J.T., and L.W. conceived of the research and provided hand labels for the shared images. N.J. coded and performed all analyses on NSD data. A.W. provided code. L.W., M.J.T. and M.M.H. designed the localizer. L.W. and M.J.T. performed and analyzed the localizer experiment. N.J., A.W., M.M.H., R.L., J.S.P., M.J.T., and L.W. wrote and edited the paper.

## Competing interests

The authors declare no competing interests.
