## [Peer Review File · Communications Biology]

Reviewers' comments:

Reviewer #1 (Remarks to the Author):

In this manuscript, the authors present their case for a novel region of the ventral visual cortex that exhibits category selectivity for food. Across several different analytic methods and a replication experiment, they find subject-specific patches of ventral cortex that are adjacent to the FFA but appear to respond selectively to food. In examining the results, I was repeatedly struck by how the food-selective region appears to straddle previously identified category selective regions (FFA, PPA, EBA, etc.). Overall, I think the approach and methods are excellent and appreciated the clarity of the writing. I have only minor comments.

1) I struggled to understand why, in Experiment 1, food images containing people, animals and faces were included. Where there not sufficient food images without people/faces? This comes up again in the PCA decomposition of these responses, where the presence of people/sociality of the food image appears to be represented in two of the three PCs.

2) Following from the above, the PCA analysis, while interesting, seems less informative given the inclusion of people, as two of the three PCs track some aspect of faces/sociality. I would be curious what would happen if only pure food images were analyzed. Similarly, the manuscript discusses at several points how the food selective region is not driven by low-level features, but the PC analysis would suggest that non-food dimensions such as prominence/relative size of food (PC1), place of food (PC2), sociality (PC3) really carry the representations. I see this as some cause for concern, as if I was to be a skeptic of this finding. I would find it curious that the "representational space" of the food images is a mix of what you might expect of the FFA (faces, sociality, animacy) and the PPA (size, foreground/background, locations). That said, experiment 2 puts some of these concerns to rest, but doesn't permit an analysis of the representational space. Long story short, I wonder if the PCA analysis, as is currently implemented with all the stimuli, is simply not that informative given the inclusion of faces/places etc. It may be worth considering removing it altogether.

Reviewer #2 (Remarks to the Author):

Jain et al provide a very rich and meticulously worked out manuscript providing evidence for neural food selectivity in the human ventral cortex, located next to functionally defined face selectivity in the ventral visual cortex. The authors followed two main research approaches. First, they used the "big data" NSD dataset of 8 participants viewing a large natural image dataset and applied a GLM-based analysis to identify responses correlating with food images. Laudibly, the authors performed their own labelling on the images also to account for alternative accounts such as reachable space, and supplemented the analysis with PCA and DNN projections of the neural space into the image space to check for particularities and groupings of images. Second, the authors designed their own food localizer which they integrated into existing localizer to confirm their NSD finding using a hypothesis-driven true experimental approach. Both approaches led to consistent results. In both introduction and discussion the authors additionally go to some length to discuss reasons for higher individual variability of food responses compared to other functional selectivities in current and past approaches. The paper is accessibly written, the methods are state-of-the art and well documented, and the claims are supported by the results. It is apparent that the manuscript went through thorough revisions among the authors, as analyses and text are deep, polished, and well thought out. I see no major flaw or obvious missing control analysis.

I hence have only minor points whose consideration may improve the manuscript.

1. The introduction is extremely long (about 2k words). I suggest shortening by moving/merging some parts to the discussion – some parts between both sections appear somewhat redundant (especially reasons for lack of prior findings), and hence can be merged to the discussion. While findings or

shortcomings of prior studies are obviously important motivators, the more detailed dissection of underlying reasons can well be moved to the discussion.

2. Main Text: Expt 1, PCA analysis: I like it when the results section can be understood at an intuitive level without necessarily reading in-depth descriptions in the methods section. For the PCA analysis, it would take only a few more words to clarify along which dimensions PCA was performed (and that this was done for ROIs).

3. Main text: Experiment 2: the main text does not refer to the N of participants, but states that 3 participants had overlapping responses – please also state here the total N to provide the necessary context.

Signed review
Andreas Bartels

Reviewer #3 (Remarks to the Author):

Rev of Jain et al COMMSBIO-22-3357-T

This is an important and clearly written paper showing selective responses to images of food in the ventral visual pathway in humans in a recently released large public fMRI database. Although two other studies reported very similar findings from the same dataset at more or less the same time, the fact that these independent analyses came up with such similar results is something to celebrate given the widely noted replication crisis in our field. Further, this study adds important new findings, showing food selectivity in four new participants for greyscale images of food with no background, thus a) replicating the food selectivity in new participants, and b) showing that food selectivity cannot be accounted for in terms of color preferences or background image confounds. This paper makes a valuable contribution and deserves publication in *Communication Biology*. It will also be a useful service to the field to make the localizer for food selectivity publicly available, as the authors promise. I have just two comments I would like to see the authors respond to, and the others they can consider but do with as they see fit:

Main Points:

1. My main critique of the paper is that it presents no information about the magnitude of response of the food regions/voxels to food and nonfood images, it only reports t values for the contrast of food > nonfood, plus the top ten images that produced the strongest response in those voxels in held-out data from each of the participants in Study 1. Given the enormous amount of data in each participant in Experiment 1 and resulting massive statistical power, the observed food selectivity could in principle reflect a rather weak difference in response between food and nonfood conditions. I would like to simply see the magnitude of response to food and to nonfood in the held-out data in these participants, and ideally also some kind of measure of selectivity (for example a correlation with food labels, or the standard measure of $(\text{preferred} - \text{nonpreferred}) / (\text{preferred} + \text{nonpreferred})$), ideally with comparable measures of selectivity of face voxels for faces and place voxels for places. Classical definitions of visual selectivity used a criteria of a 2:1 response ratio of preferred: nonpreferred stimuli. While that criterion is not set in stone, it would be useful to know if the current findings are in that ballpark. This would help test whether the selectivity for food is of similar strength to the previously reported category selectivities for faces and places etc. The magnitude of food selectivity is also relevant because our paper (Khosla et al 2022) found very strong selectivity using a matrix factorization method, and lower selectivity for voxelwise responses.

2. The statistics for Experiment 2 are handled in a somewhat informal fashion. For Experiment 2, the text simply says that: "The results of the localizer replicate the findings of Experiment 1" and the figure shows 1-sided t maps in each of the four participants, directing us to Figure S8 for significance

thresholds (which are pretty lax at FDR 0.05, especially when only four participants are included). This is pretty non standard for current neuroimaging practice. I would recommend making clearer statistic claims in the main text and figures of the paper.

Other Points

1. The conclusion from the PCA analysis of Experiment 1 is that "PCA can be used to uncover both the large-scale topography within food-selective areas and the interaction of food coding with other semantic dimensions. Namely, we find that the representation of food in the food regions appears to be organized in gradients across cortex that relate food to other important information processed nearby (social and place-related information)." But my guess is that food selectivity simply overlaps a little bit with face selectivity in some voxels, and with scene selectivity in others, and the PCA is simply pulling out these different dimensions that are already known in the ventral pathway. So these results need not show anything other than which selectivities are adjacent to (and sometimes slightly overlapping with) food selectivity, which was already clear from other analyses. The same applies to the clustering analyses. But "the interaction of food coding with other semantic dimensions" sounds like the authors are making claims beyond this. It would be great to clarify whether the authors think these data show something beyond the topographic adjacencies of food selectivity with other selectivities in the cortex, and if so what that is.

2. Several topics covered in this paper were discussed in some detail in our published paper, but are not cited in that context. I am not insisting on being cited, I think these are judgement calls, I am just flagging these things for the authors and editor to consider:

a. Much discussion in Jain et al was devoted to the question of why food selectivity was not found previously. Our published paper (Khosla et al 2022) also discussed this same question at length, and offered a hypothesis, and evidence for that hypothesis: that food-selective neural populations are more spatially intermixed with other neural populations that are selective populations for faces, places, etc, so food selectivity suffers disproportionately when measured in pure voxel responses (rather than matrix components which can de-mix those responses). Yet our hypothesis and our evidence for it (e.g. in Figure 6 of our paper) was not mentioned.

b. Like Jain et al, our paper discussed the anatomical variability across participants of the food component in Figure S5 and in the text: "To quantify the inter-subject variability in the anatomy of Component 3 and compare it with other components (Figure S5), we further measured the correlation of the MNI-registered weight map for each participant with the average weight map across the 7 other participants (averaged across 8 folds). This analysis showed the highest inter-subject correlation of the weight maps for Component 1 (scenes), followed by Components 2 and 4 (faces and text, respectively), and then Components 3 (food) and 5 (bodies)".

c. Our paper made specific predictions based on our encoding model of the responses of the food component to greyscale images (see Figure 4c). It is cool to see those model predictions borne out by Experiment 2 in Jain et al, yet our predictions are not mentioned.

Referees' comments: This document includes all referees' comments as communicated. For clarity, we have numbered each comment as **REFeree#.COMMENT#**. Our point-by-point actions/replies to each comment immediately follow the comment (and are marked through bullet points and indentation). As per journal guidelines we also include the new figures that were added in response to specific comments. We have strived to thoroughly address each comment, often implementing significant changes in our manuscript. We appreciate the thoughtfulness of the comments in the ways in which they have helped us improve our paper.

Sincerely, Leila Wehbe and Michael Tarr (for all authors)

Referee #1 (Remarks to the Author):

In this manuscript, the authors present their case for a novel region of the ventral visual cortex that exhibits category selectivity for food. Across several different analytic methods and a replication experiment, they find subject-specific patches of ventral cortex that are adjacent to the FFA but appear to respond selectively to food. In examining the results, I was repeatedly struck by how the food-selective region appears to straddle previously identified category selective regions (FFA, PPA, EBA, etc.). Overall, I think the approach and methods are excellent and appreciated the clarity of the writing. I have only minor comments.

1.1 I struggled to understand why, in Experiment 1, food images containing people, animals and faces were included. Where there not sufficient food images without people/faces? This comes up again in the PCA decomposition of these responses, where the presence of people/sociality of the food image appears to be represented in two of the three PCs.

- We appreciate the referee's comment here, however there are several reasons we feel that using natural images (which is a trend in the wider field - a major point of NSD is the desirability of using more complex, real-world images) is both desirable and does not dilute our findings. First, some critiques of category-selective responses in the visual cortex have argued that using isolated objects against blank backgrounds actually overstates the degree of selectivity. Common visual processes such as segmentation and spatial attention are functionally diminished under such conditions and there have been concerns that differences in the level of responses between categories might be partially a ceiling effect - how much processing can be deployed with respect to a given category when there is no competing information. Thus, obtaining category-selective responses with more complex, natural images is often considered a stronger test (i.e., greater attention load and competition, the need to perform segmentation, etc.; some localizer studies have trended this way). Note also that our analysis (in Fig.2) which directly compared food selective areas with face-selective areas provides a control for this

concern. That is, we are contrasting responses related to food images with responses related to images with people. Our analysis found these areas to be non-overlapping. This eliminates specific potential confounds related to faces/people in food images. Second, we did a variety of analyses on subgroups of food images and found no differences in the qualitative nature of our results. That is, we used several different visual clustering algorithms to try and identify visually separable subgroups of food images (which ends up including closeups of food alone). However, none of these analyses produced any difference in the overall pattern of results (although due to less images in each subgroup, the power was lower). As such, we are confident that our results relate to the appearance of food in the image regardless as to whether the food is alone or in context. Third, we note in our paper and in the cited, parallel, Pennock and Khosla papers, that including real-world context might have actually helped in identifying food selectivity in all three of our papers (and in the BOLD5000 dataset as well) (of course our Experiment 2 suggests that this is not the only factor and that by considering individual subjects separately, it is likewise possible to identify food selectivity using food alone).

- Similarly, the Khosla et al. (2022) paper using the same dataset obtained similar findings regarding food selectivity, but, critically, also replicated those findings in BOLD5000. The food images in BOLD5000 are drawn in part from ImageNet and included approximately 50 food categories, all of which are typically food objects alone (i.e., “pure food”). This replication, along with the results of Experiment 2, suggest that the presence of faces, people etc. is not a factor in the basic finding of food selectivity. (see below for the new discussion of PCA we include in the paper to address these concerns)
- In terms of developing a more fine-grained understanding of how category-selective regions are organized, using food in real-world contexts makes sense. Note that there is already a growing literature on partitioning scene-selective regions into different functional subregions that capture different aspects of scene representation (and FFA has been functionally partitioned into FFA1 and FFA2). Since food is strongly associated with social activities, food with people and in other appropriate contexts helps us explore similar partitioning of food-selective regions into functional subregions. Thus, similar to our point above, if anything, prior studies of categories in isolation may miss the dimensionality of any category-selective representations - are they really just *category X qua category X* or are they more complex representations of different functional roles for category X? Thus, not only do we take our results as helping elucidate a better understanding of how the neural representation of food is organized, but our results hint that more complex, contextualized images should be used to study face, body, tool, etc. regions to similarly explore the potential underlying dimensions.

1.2 Following from the above, the PCA analysis, while interesting, seems less informative given the inclusion of people, as two of the three PCs track some aspect of faces/sociality. I would be curious what would happen if only pure food images were analyzed. Similarly, the manuscript

discusses at several points how the food selective region is not driven by low-level features, but the PC analysis would suggest that non-food dimensions such as prominence/relative size of food (PC1), place of food (PC2), sociality (PC3) really carry the representations. I see this as some cause for concern, as if I was to be a skeptic of this finding. I would find it curious that the “representational space” of the food images is a mix of what you might expect of the FFA (faces, sociality, animacy) and the PPA (size, foreground/background, locations). That said, experiment 2 puts some of these concerns to rest, but doesn’t permit an analysis of the representational space. Long story short, I wonder if the PCA analysis, as is currently implemented with all the stimuli, is simply not that informative given the inclusion of faces/places etc. It may be worth considering removing it altogether.

- With respect to the referee’s concern that “the “representational space” of the food images is a mix of what you might expect of the FFA (faces, sociality, animacy) and the PPA (size, foreground/background, locations).” We suggest that the spatial localization of food selectivity actually reinforces the idea that food responses do leverage adjacency to the FFA and PPA. As discussed in our paper and many prior papers (as well as Pennock’s and Khosla’s papers, there is no apparent visual basis for food selectivity. As such, it must emerge as a result of interactions with other systems - reward, gustatory, social processing etc. What we might expect (and as studied in papers focusing more on hunger state or BMI) is that food responses are aggregation of a wide variety of factors, including faces and place associations.
- In light of the above arguments we respectfully disagree with the reviewer about removing our PCA analysis. First, we see the PCA as actually informing more complex representations than as have been articulated in the past. Second, food is an ecologically important category, but food selectivity almost surely arises out of interactions with other roles for food - reward, social context, satiation, preference etc. - so the kind of dimensionality we observe may be intrinsic to food representation (and may arise out of the mechanisms whereby food selectivity emerges in the first place). In sum, we feel that our PCA results, while clearly not definitive, provide information that is useful to those interested in the neural representations of food. Not only does it highlight some of the possible dimensions of food representation, but it suggests other, fruitful future directions for finer-grained studies of food representation. Moreover, it may also provide a roadmap for studies of other domains of category selectivity that have likewise been associated with different functional dimensions, but have not, as yet been strongly fleshed out
- At the same time, we acknowledge the referee’s concerns (as well as those of Referee #3) regarding our interpretation of our PCA results. To address those concerns head on, we have added the following to our discussion regarding PCA:
Lines 250-272: “One concern with this interpretation is that the presence of people and/or faces in conjunction with food within our test images may lead to responses driven more by these well-known category-selective domains rather than food \textit{per se}. That is,

the PCs we obtain may be a result of the PCA picking up on food selectivity that overlaps with adjacent face- and body-selective regions. However, as demonstrated in Experiment 2, we obtain the same pattern of food selectivity using “pure” food images. Similarly, Khosla et al. (2022) replicated their results (and our’s) using BOLD5000 \cite{chang2019bold5000} - a different dataset that includes primarily food alone images (images in BOLD5000 are drawn in part from ImageNet and include approximately 50 food categories). These results suggest that the presence of faces, people etc. is not a factor in the basic finding of food selectivity. From a theoretical standpoint, we posit that the organization of food responses might reflect more than the adjacencies of food selectivity with other category-selective domains. Rather, the visual heterogeneity of food implicates non-visual factors as the underlying drivers in food-related visual responses. Based on prior work (e.g., \cite{CHEN201620,Huerta2014}) and the adjacency of food-selective regions to the FFA and PPA, we propose that food responses are the aggregation of a wide variety of factors, including explicit social (face and body) and place associations, reward feedback circuits, and gustatory signals, all of which play some functional role in food processing as realized in the ventral visual cortex. Future work should explore this hypothesis, examining how the interaction of food with other systems gives rise to visual food selectivity.”

Lines 289-293: “The lack of correspondence in our clustering results suggests that the responses in food-selective areas do not organize easily into clusters based on visual similarity, scene semantics, or object semantics (and reinforces the complex, high-level nature of the dimensions found using PCA).”

Referee #2 (Remarks to the Author):

Jain et al provide a very rich and meticulously worked out manuscript providing evidence for neural food selectivity in the human ventral cortex, located next to functionally defined face selectivity in the ventral visual cortex. The authors followed two main research approaches. First, they used the “big data” NSD dataset of 8 participants viewing a large natural image dataset and applied a GLM-based analysis to identify responses correlating with food images. Laudibly, the authors performed their own labelling on the images also to account for alternative accounts such as reachable space, and supplemented the analysis with PCA and DNN projections of the neural space into the image space to check for particularities and groupings of images. Second, the authors designed their own food localizer which they integrated into existing localizer to confirm their NSD finding using a hypothesis-driven true experimental approach. Both approaches led to consistent results. In both introduction and discussion the authors additionally go to some length to discuss reasons for higher individual variability of food responses compared to other functional selectivities in current and past approaches.

The paper is accessibly written, the methods are state-of-the art and well documented, and the claims are supported by the results. It is apparent that the manuscript went through thorough revisions among the authors, as analyses and text are deep, polished, and well thought out. I see no major flaw or obvious missing control analysis.

I hence have only minor points whose consideration may improve the manuscript.

2.1 The introduction is extremely long (about 2k words). I suggest shortening by moving/merging some parts to the discussion – some parts between both sections appear somewhat redundant (especially reasons for lack of prior findings), and hence can be merged to the discussion. While findings or shortcomings of prior studies are obviously important motivators, the more detailed dissection of underlying reasons can well be moved to the discussion.

- As suggested, appropriate parts (and a significant part) of the introduction have been moved/merged to the discussion section - we agree that this streamlines the paper. We also merged/compressed several paragraphs in the introduction and discussion for clarity. This involves Lines x53-59, 72-76, and Lines 482-556.

2.2 Main Text: Expt 1, PCA analysis: I like it when the results section can be understood at an intuitive level without necessarily reading in-depth descriptions in the methods section. For the PCA analysis, it would take only a few more words to clarify along which dimensions PCA was performed (and that this was done for ROIs).

- We have added in text clarifying our PCA results, adding in text explaining which voxels were included in our analyses in the results section
Lines 220-226: “PCA was run using only voxels within the identified food region (Fig.~\ref{fig:food-voxels}) across all subjects crossed with the neural activity associated with the shared food images. This PCA produces for each voxel a set of principal component scores that capture the projection of its high-dimensional response profile across all images onto a lower dimensional subspace. The axes of this subspace -- shared semantic axes -- correspond to the dimensions in food image space that are most strongly reflected in the voxel responses (Fig.~\ref{fig:PCA}A).”

2.3 Main text: Experiment 2: the main text does not refer to the N of participants, but states that 3 participants had overlapping responses – please also state here the total N to provide the necessary context.

- We apologize for the omission and now specify the N of participants for Experiment 2 in the main text:
Lines 298-300: “Next, we designed a standard food ``localizer" and collected new fMRI data in 4 new subjects to test whether we could replicate our results in a controlled experiment.”

Line 336: “at most 3 of 4 subjects have a significant contrast in the same region of the left fusiform”

Signed review
Andreas Bartels

Referee #3 (Remarks to the Author):

Rev of Jain et al COMMSBIO-22-3357-T

This is an important and clearly written paper showing selective responses to images of food in the ventral visual pathway in humans in a recently released large public fMRI database. Although two other studies reported very similar findings from the same dataset at more or less the same time, the fact that these independent analyses came up with such similar results is something to celebrate given the widely noted replication crisis in our field. Further, this study adds important new findings, showing food selectivity in four new participants for greyscale images of food with no background, thus a) replicating the food selectivity in new participants, and b) showing that food selectivity cannot be accounted for in terms of color preferences or background image confounds. This paper makes a valuable contribution and deserves publication in *Communication Biology*. It will also be a useful service to the field to make the localizer for food selectivity publicly available, as the authors promise. I have just two comments I would like to see the authors respond to, and the others they can consider but do with as they see fit:

Main Points:

3.1 My main critique of the paper is that it presents no information about the magnitude of response of the food regions/voxels to food and nonfood images, it only reports t values for the contrast of food > nonfood, plus the top ten images that produced the strongest response in those voxels in held-out data from each of the participants in Study 1. Given the enormous amount of data in each participant in Experiment 1 and resulting massive statistical power, the observed food selectivity could in principle reflect a rather weak difference in response between food and nonfood conditions. I would like to simply see the magnitude of response to food and to nonfood in the held-out data in these participants, and ideally also some kind of measure of selectivity (for example a correlation with food labels, or the standard measure of (preferred - nonpreferred)/(preferred + nonpreferred), ideally with comparable measures of selectivity of face voxels for faces and place voxels for places. Classical definitions of visual selectivity used a criteria of a 2:1 response ratio of preferred: nonpreferred stimuli. While that criterion is not set in stone, it would be useful to know if the current findings are in that ballpark. This would held test

whether the selectivity for food is of similar strength to the previously reported category selectivities for faces and places etc. The magnitude of food selectivity is also relevant because our paper (Khosla et al 2022) found very strong selectivity using a matrix factorization method, and lower selectivity for voxelwise responses.

- Thank you for this suggestion. We have now computed the measure of $(\text{preferred} - \text{non-preferred})/(\text{preferred} + \text{non-preferred})$ for both datasets. To evaluate the non-preferred activation, we chose the maximum of the nonpreferred categories instead of the mean, which is a much more stringent baseline.
- A criteria of a response ratio of preferred: non-preferred stimuli being at least 2:1 would correspond to the $(\text{preferred} - \text{non-preferred})/(\text{preferred} + \text{non-preferred})$ being above $1/3$.
- The results using the localizers show a clearly large selectivity measure for food, on par with faces and places (see the new Supplementary Figure S11).
- The results using the NSD data are a little more variable across subjects (see the new Supplementary Figure S10). For some subjects, the strength of selectivity for food is the same as faces, for some it's more and for some it's less. This variability might be due to the fact that the max statistic is more variable than the mean statistic, and this is more pronounced in the NSD dataset since different categories have different frequencies (in contrast with the localizer dataset in which the categories are balanced).
- We have added references to these results in the appropriate results sections at Lines 177-189 and Lines 309-324.
- The new figures are as follows:

Supplementary Figure S10. Experiment 1. Voxel-wise selectivity for food and faces viewed in natural scene images. Selectivity was defined as: $\frac{\text{preferred} - \text{non-preferred}}{|\text{preferred}| + |\text{non-preferred}|}$, where the non-preferred baseline activity is the maximum activity related to other categories. To measure selectivity for Category c , we compute $\frac{\beta_c - \max_{i \neq c} \beta_i}{|\beta_c| + \max_{i \neq c} \beta_i}$, where β_i is the weight of the OLS encoding model corresponding to Category i .

Supplementary Figure S11. Experiment 2. Voxel-wise selectivity for food, faces and places viewed in the localizer images. Selectivity was defined as: $\frac{\text{preferred} - \text{non-preferred}}{|\text{preferred}| + |\text{non-preferred}|}$, where the non-preferred baseline activity is the maximum activity related to other categories. To measure selectivity for Category c , we compute $\frac{\beta_c - \max_{i \neq c} \beta_i}{|\beta_c| + \max_{i \neq c} \beta_i}$, where β_i is the weight of the OLS encoding model corresponding to Category i .

3.2 The statistics for Experiment 2 are handled in a somewhat informal fashion. For Experiment 2, the text simply says that: "The results of the localizer replicate the findings of Experiment 1" and the figure shows 1-sided t maps in each of the four participants, directing us to Figure S8 for significance thresholds (which are pretty lax at FDR 0.05, especially when only four participants are included). This is pretty non standard for current neuroimaging practice. I would recommend making clearer statistic claims in the main text and figures of the paper.

- Thank you for this comment, we realize that our description was different from typical practice. We have changed our significance threshold to be more restrictive ($\alpha = 0.001$).
Lines 309-315.
- Our setup is different from usual localizer studies because we don't use a group level analysis, but instead we replicate the findings in each of the subjects. And we find the results are repeatable in each subject. We included these points in the results section and have added additional statistical details at the appropriate points.

Other Points

3.3 The conclusion from the PCA analysis of Experiment 1 is that "PCA can be used to uncover both the large-scale topography within food-selective areas and the interaction of food coding with other semantic dimensions. Namely, we find that the representation of food in the food regions appears to be organized in gradients across cortex that relate food to other important information processed nearby (social and place-related information)." But my guess is that food selectivity simply overlaps a little bit with face selectivity in some voxels, and with scene selectivity in others, and the PCA is simply pulling out these different dimensions that are already known in the ventral pathway. So these results need not show anything other than which selectivities are adjacent to (and sometimes slightly overlapping with) food selectivity, which was already clear from other analyses. The same applies to the clustering analyses. But "the interaction of food coding with other semantic dimensions" sounds like the authors are making claims beyond this. It would be great to clarify whether the authors think these data show something beyond the topographic adjacencies of food selectivity with other selectivities in the cortex, and if so what that is.

- See extensive discussion of this and related questions in response to Referee #1. More to the point, we have added the following paragraph when presenting our PCA results (Lines 261-272): "From a theoretical standpoint, we posit that the organization of food responses might reflect more than the adjacencies of food selectivity with other category-selective domains. Rather, the visual heterogeneity of food implicates non-visual factors as the underlying drivers in food-related visual responses. Based on prior work (e.g., \cite{CHEN201620,Huerta2014}) and the adjacency of food-selective regions to the FFA and PPA, we propose that food responses are the aggregation of a wide variety of factors, including explicit social (face and body) and place associations, reward feedback circuits, and gustatory signals, all of which play some functional role in food processing as realized in the ventral visual cortex. Future work should explore this hypothesis, examining how the interaction of food with other systems gives rise to visual food selectivity."

3.4 Several topics covered in this paper were discussed in some detail in our published paper, but are not cited in that context. I am not insisting on being cited, I think these are judgement calls, I am just flagging these things for the authors and editor to consider:

a. Much discussion in Jain et al was devoted to the question of why food selectivity was not found previously. Our published paper (Khosla et al 2022) also discussed this same question at length, and offered a hypothesis, and evidence for that hypothesis: that food-selective neural populations are more spatially intermixed with other neural populations that are selective populations for faces, places, etc, so food selectivity suffers disproportionately when measured in pure voxel responses (rather than matrix components which can de-mix those responses). Yet our hypothesis and our evidence for it (e.g. in Figure 6 of our paper) was not mentioned.

- This hypothesis is now mentioned in our discussion, particularly where we discuss the spatial distribution of food-selective responses
Lines 470-474: “Consistent with this interpretation and the results of our Experiment 2, Adamson and Troiani~\cite{adamson2018} report separation in the peak coordinates for face and food clusters for individual subjects. Similarly, Khosla~et.~al. \cite{khosla2022} found that the food-selective component of their results was less spatially correlated across subjects as compared to other category-selective components.”

3.5 Like Jain et al, our paper discussed the anatomical variability across participants of the food component in Figure S5 and in the text: "To quantify the inter-subject variability in the anatomy of Component 3 and compare it with other components (Figure S5), we further measured the correlation of the MNI-registered weight map for each participant with the average weight map across the 7 other participants (averaged across 8 folds). This analysis showed the highest inter-subject correlation of the weight maps for Component 1 (scenes), followed by Components 2 and 4 (faces and text, respectively), and then Components 3 (food) and 5 (bodies)".

- Thank you for pointing this out. This finding is now mentioned when discussing the spatial variability of food-selective responses across subjects in both our results and discussion
Lines 330-333: “we see less spatial agreement among subjects in the food vs. other contrast as compared to the face, body, and place contrasts (face vs. all, body vs. all, place vs. all, and words vs. all; see Suppl. Fig.~\ref{fig:localizer-all}; in parallel work, \cite{khosla2022} observe a similar pattern as illustrated in Fig.~S5 of their paper).”

3.6 Our paper made specific predictions based on our encoding model of the responses of the food component to greyscale images (see Figure 4c). It is cool to see those model predictions borne out by Experiment 2 in Jain et al, yet our predictions are not mentioned.

Nancy Kanwisher

- This prediction and its confirmation by the results of Experiment 2 are now mentioned when discussing the results of Experiment 2
Lines 321-324: “Our new results also indicate that factors such as color or food appearing in a natural scene are not essential for obtaining selective activation for food (in parallel work, Khosla et.~al. \cite{khosla2022} likewise predicted that food-selective responses should be obtained using grayscale images; see their Fig.~4c).”

Reviewers' comments:

Reviewer #1 (Remarks to the Author):

I've reviewed the authors manuscript and letter. I still think this is an excellent study, although I continue to quibble about certain aspects. Let's get right to it.

1.1. In my initial review, I indicated that I had some concern over the inclusion of images with people in the first experiment. Particularly as people and sociality appear in 2 of the 3 PCs from the PCA analysis. The authors rebuttal discusses the (a) importance of using naturalistic images, (b) other people found similar regions for food, (c) category selectivity might benefit by using stimuli that contain multiple categories.

I don't necessarily disagree with these points, but this study is a specific case of determining whether a category selective region exists and, given the way the paper is written and the outlet the authors chose, it suggests that this will be THE paper establishing a food selective region in VT cortex. Thus, in this case, I maintain a more reductive approach that firmly establishes category selectivity and does away with all potential confounds is necessary. If the first FFA paper threw in images of faces in scenes with all kinds of other categories and made these same claims, I don't think it would have had the same impact.

So why not run it again with just food images to preempt any future naysayers?. Based on figure 1, it seems like the presence of people in the image set is relatively uncommon and therefore there should be ample images to work with.

1.2 I stated I didn't see much value in the PCA analysis as it was picking up on non-food dimensions in the image set (which any other localizer type task would have excluded). So in some sense, the PCA is dependent on this particular image set and wouldn't replicate in a pure food image set. The authors counter that people, faces, etc. are all a vital part of food representations and that food selective cortex would reflect that. Perhaps so, as are hands, tools (i.e. cutlery) and possibly a number of other dimensions that co-occur with food. The selection of just these dimensions seems more an artifact of the image set than any principled reason to include them and so, again, I don't see the value of this analysis in a paper aiming to establish a food selective patch of cortex. I don't think the PCA needs to be cut, but a PCA based on a purer set of food images would be likely be more informative. As I mentioned above, if Figure 1 is any indication, there should be plenty shared across the subjects.

The following comment is new:

The additional details on the PCA have me somewhat puzzled and led me to dig further into the methods and I wondered if the authors could clarify a few points. According to the methods, you have participants along rows of the data and voxels (features) as columns and then are "reducing along the image dimensions". Where I struggle is that this table has more than two dimensions (subjects, voxels, images). I'm familiar with running PCA on, for instance, mean features across a group of subjects (i.e, group averaged voxel responses for a set of observations). I'm familiar with multi-table PCA methods (STATIS) for combining PCAs across subjects and deriving a common solution. But neither seems to be the case here. Can you describe in more detail how exactly your performing PCA on voxel responses to images across subjects? if it's actually a separate PCA model per subject, how are you ensuring the order of PCs is equivalent when you go to average PCs as indicated in Figure 4. If these are separate PCA per subject, then PC1 for subject 1 need not correspond to PC1 for subject 2, etc. Again, it's been awhile since I've had to dig this deep into PCA so if I'm missing something obvious, then I thank you for indulging my curiosity.

Reviewer #3 (Remarks to the Author):

I think this paper is in good shape and I am happy to recommend publication.

I'll mention just one remaining concern that the authors can either address or not as they see fit.

I am not convinced by the analysis claiming that voxelwise food selectivity is on average on a par with face selectivity, for several reasons:

i) using the highest nonpreferred category chosen separately for each voxel is understandable (selectivity ratios are unstable for individual voxels and behave unstably when some response magnitudes are negative), but far from ideal. If I am understanding this right the baseline condition presumably differed across voxels as a function of which category produced the highest nonpreferred response; this makes this measure strange as it does not measure the same thing across voxels. Also, this measure is highly vulnerable to the noise inevitable in the response of individual voxels.

ii) there should have been lots of voxels with NEGATIVE selectivity using this measure but none are shown.

iii) the claim that face and food selectivity was similar based on those voxel maps is not quantified, and seems to be based on eyeballing the darkness of the red bits on figures S10 and S11. (If there is another method it should be made clear)

iv) The conclusion differs from the similar voxelwise analyses of the same data in Fig 6 in our paper:

https://web.mit.edu/bcs/nklab/media/pdfs/Khosla_CB2022.pdf

Of course it could be our quantification that was wrong :-) but I suspect not.

Given this concern the authors might consider alternative quantifications that enable the selectivity for food in the "food voxels" to be compared to selectivity for faces in the "face voxels".

Reviewers' comments:

Reviewer #1 (Remarks to the Author):

I've reviewed the authors manuscript and letter. I still think this is an excellent study, although I continue to quibble about certain aspects. Let's get right to it.

1.1. In my initial review, I indicated that I had some concern over the inclusion of images with people in the first experiment. Particularly as people and sociality appear in 2 of the 3 PCs from the PCA analysis. The authors rebuttal discusses the (a) importance of using naturalistic images, (b) other people found similar regions for food, (c) category selectivity might benefit by using stimuli that contain multiple categories.

I don't necessarily disagree with these points, but this study is a specific case of determining whether a category selective region exists and, given the way the paper is written and the outlet the authors chose, it suggests that this will be THE paper establishing a food selective region in VT cortex. Thus, in this case, I maintain a more reductive approach that firmly establishes category selectivity and does away with all potential confounds is necessary. If the first FFA paper threw in images of faces in scenes with all kinds of other categories and made these same claims, I don't think it would have had the same impact.

So why not run it again with just food images to preempt any future naysayers?. Based on figure 1, it seems like the presence of people in the image set is relatively uncommon and therefore there should be ample images to work with.

While we understand the Reviewer's point of view, it would seem to be a larger concern if we did not include Experiment 2 - which is an entirely "pure food" localizer that directly establishes food selectivity in VT cortex and results consistent with Experiment 1. So we suggest that our results *in toto* can be thought of as "firmly" establishing food selectivity with a "reductive" approach as suggested by the Reviewer. This point is strengthened further by the new selectivity analyses as suggested by Reviewer #3 (Figure S12 illustrates similar selectivity for food, faces, and places). One way to think about this is to take Experiment 2 as the "pure" case and then Experiment 1 as validating food selectivity for complex, real-world images (something that most work on category selectivity never does - and which our lab is now pursuing more broadly across categories as a followup scientific question). To try and illustrate our point here, wouldn't THE paper on face selectivity (e.g., the original Sergent paper or the later Kanwisher paper) both have been stronger if they had included not only the standard "pure" face localizer they presented (in PET and fMRI respectively), but also included a followup validation study in which "faces in the wild" were also demonstrated to give rise to a similar pattern of face selectivity?

1.2 I stated I didn't see much value in the PCA analysis as it was picking up on non-food dimensions in the image set (which any other localizer type task would have excluded). So in some sense, the PCA is dependent on this particular image set and wouldn't replicate in a pure food image set. The authors counter that people, faces, etc. are all a vital part of food representations and that food selective cortex would reflect that. Perhaps so, as are hands, tools (i.e. cutlery) and possibly a number of other dimensions that co-occur with food. The selection of just these dimensions seems more an artifact of the image set than any principled reason to include them and so, again, I don't see the value of this analysis in a paper aiming to establish a food selective patch of cortex. I don't think the PCA needs to be cut, but a PCA based on a purer set of food images would be likely be more informative. As I mentioned above, if Figure 1 is any indication, there should be plenty shared across the subjects.

With respect to the PCA, the Reviewer comments: "I don't think the PCA needs to be cut, but a PCA based on a purer set of food images would be likely be more informative". This is an interesting and fair question for which we have two thoughts.

1. By definition, the PCA cannot be informative as for how food representations might interact with other categories if those categories are omitted from the PCA. We couldn't learn anything about social/people interactions if no pictures with food and people (in social contexts or alone) are not included. So we certainly would like to push back that such "interactive" analyses across complex natural scenes with multiple categories are important - we can't understand or test what we don't include as a possibility to look for in our studies. Indeed, one could argue that most "category-selective" studies presuppose a particular model of neural representation in which domains are "pure" and then create studies that only test for that model by failing to including conditions (e.g., single objects against decontextualized backgrounds) and/or analyses (e.g., subtractive comparisons only) that do not allow for more complex "interactive" representations in VT (see the recent excellent work by Konkle on gradients in VT).
2. At the same time, we acknowledge the Reviewer's point that a "pure food" PCA might carry different kinds of information than the PCA we include. It would remove any confounds regarding people and we might learn something about the dimensionality of food alone (which we presume would be the point of running a PCA with pure food pictures). It is possible there are no finer grained food alone organizing principles, but perhaps some factor such as calories, food appeal, sweet/salty/sour/spicy/bitter, prepared/raw, food groups, etc would come out as dimensions of the PCA? Such an analysis might be informative as to different

possible dimensions of food *qua* food representation. We suggest that the original food-in-context PCA with all pictures is worth keeping, but that the potentially informative food with human faces and human bodies removed PCA is also worth including in that it might tell us whether our results for food selectivity are stable for food images only. Alternatively, it might tell us something about how food itself is organized within food-selective regions.

As such, we ran the PCA with food images excluding all images that contained either a human face or a human body (a handful of images containing an animal were not removed). The results are shown in Supplementary Figure S7 (included below). One can see that the brain maps for Figure S7A are *remarkably similar* to the brain maps in Figure 4A. This strongly suggests that a “pure” food analysis does not change the results regarding the spatial distribution of food selectivity in the ventral visual cortex. Moreover, PC1 and PC2 appear to capture the same dimensions across the full PCA and the pure food PCA. However, PC3 from the full PCA captured social aspects interacting with food - that dimension is missing from the pure food PC3 as it should be since there are no longer any images in the analysis carrying such social information. But to reiterate the main finding here, the pure food PCA produces patterns on the brain that are nearly identical to those for the original, full PCA. Thus, along with the results of Experiment 2, we are confident that our results reflect strong food selectivity as an organizing principle of VT. We thank the reviewer for raising this issue because we agree with them that the inclusion of the new PCA as reflected in Figure S7 strengthens our paper.

Supplementary Figure S7. Experiment 1. PCA of responses from food-selective regions excluding images containing human faces and human bodies. (A) Average principal component score across subjects for PC1, PC2, and PC3, shown on the MNI surface. Blue-green indicates high, brown indicates low PC scores. In (B) we plot the top and bottom images for PC1, PC2, and PC3 along a linear axis (lowest to highest from left to right). We include the 4 lowest and 4 highest images for ease of visualization. The patterns across the brain that emerge here are remarkably well aligned with the patterns seen for the full PCA (Figure 4A). Qualitatively, PC1 and PC2 again seem to distinguish large-scale images of food-related places from close-by images of food, as well as capturing the prominence of food in an image, separating images with focus on food in the foreground from those with food in the background. PC3, however, appears less interpretable and does not support any useful inferences.

The following comment is new:

The additional details on the PCA have me somewhat puzzled and led me to dig further into the methods and I wondered if the authors could clarify a few points. According to the methods, you have participants along rows of the data and voxels (features) as columns and then are "reducing along the image dimensions". Where I struggle is that this table has more than two dimensions (subjects, voxels, images). I'm familiar with running PCA on, for instance, mean features across a group of subjects (i.e, group averaged voxel responses for a set of observations). I'm familiar with multi-table PCA methods (STATIS) for combining PCAs across subjects and deriving a common solution. But neither seems to be the case here. Can you describe in more detail how exactly your performing PCA on voxel responses to images across subjects? if it's

actually a separate PCA model per subject, how are you ensuring the order of PCs is equivalent when you go to average PCs as indicated in Figure 4. If these are separate PCA per subject, then PC1 for subject 1 need not correspond to PC1 for subject 2, etc. Again, it's been awhile since I've had to dig this deep into PCA so if I'm missing something obvious, then I thank you for indulging my curiosity.

Thank you for pointing out this omission. The matrix we ran PCA on is indeed two dimensions (# of voxels by # of images). The number of rows in this case is all previously identified food selective voxels from all subjects concatenated together. (e.g. if we have 200 food voxels from each subject, then the matrix (across 4 subjects) will have 800 rows. With this method, we only have one PCA model for all subjects.

The relevant Methods text now reads: *Then, we ran PCA on a matrix of concatenated 'food-relevant' voxels for all subjects (rows) by the activity related to shared food images (columns), reducing along the image dimension (the columns). That is, the number of rows consists of all previously identified food selective voxels from all subjects concatenated together (i.e., if we have 200 food voxels for each subject, then the matrix across 4 subjects will have 800 rows). Thus, the matrix on which we ran PCA is two dimensional (# of voxels by # of images) and we have only one PCA model for all subjects.*

Reviewer #3 (Remarks to the Author):

I think this paper is in good shape and I am happy to recommend publication. I'll mention just one remaining concern that the authors can either address or not as they see fit.

I am not convinced by the analysis claiming that voxelwise food selectivity is on average on a par with face selectivity, for several reasons:

i) using the highest nonpreferred category chosen separately for each voxel is understandable (selectivity ratios are unstable for individual voxels and behave unstably when some response magnitudes are negative), but far from ideal. If I am understanding this right the baseline condition presumably differed across voxels as a function of which category produced the highest nonpreferred response; this makes this measure strange as it does not measure the same thing across voxels. Also, this measure is highly vulnerable to the noise inevitable in the response of individual voxels.

Using the highest nonpreferred category chosen separately for each voxel is actually more *conservative* and less vulnerable to noise. That is, the Reviewer seems to be suggesting that a common baseline category response is a better comparison to food responses to measure food selectivity. But that baseline is guaranteed to be lower on average than using the highest nonpreferred category as baseline (because we used the *highest* possible). So food selectivity can only look stronger using a fixed baseline, which seems scientifically less transparent and may misrepresent the strength of selectivity for food (making it look stronger than it is). So the highest nonpreferred response seems to be the fairest and most conservative. Moreover, it removes the assumption that voxels are uniform in their selectivity in a given brain region - it is equally possible that voxel preference might vary voxel by voxel and not region by region (a hypothesis promoted by multiple groups). So fixing the baseline presumes there is some spatial uniformity and biases results to look more uniform than they might actually be. We address this question in the revised manuscript in the following addition to the caption for Figure S11:

*Voxel-wise selectivity for food and faces viewed in natural scene images.} Selectivity was defined as: $\frac{\text{preferred} - \text{non-preferred}}{\text{preferred} + \text{non-preferred}}$ where the non-preferred baseline activity is the *maximum* activity related to any other category. This is a *conservative* and less biased measure of selectivity in that we consider the food or face response of each voxel relative to the highest response across all other possible categories for that voxel (rather to a single fixed baseline category). This approach allows for voxel by voxel variation in selectivity rather than assuming that an entire region's voxels respond in an uniform manner.*

ii) there should have been lots of voxels with NEGATIVE selectivity using this measure but none are shown.

Yes, the reviewer is correct, there are many negative voxels - as there should be when measuring selectivity for a category (say food) in a region for which that category is not the preferred category (say the FFA). We now clarify this by including the following explanation in the caption for Figure S11 (as well as S12):

Selective voxels for each category are plotted on inflated views of individual subjects' brains. To better visualize the largest number of category selective voxels, positive selectivity values greater than 0.5 and negative selectivity values are not plotted (because negative selectivity denotes non-preferred category-selective responses that obscure the preferred category-selective regions; e.g., when measuring food selectivity for voxels in the FFA, the highest response for a non-food category was typically for

faces and this response was typically higher than for food for that voxel, thereby producing a negative food selectivity index).

We also clarify this in the caption for Figure S12. Moreover, while negative selectivity would make interpreting the brain maps more difficult, negative selectivity values are included in histograms in Figure S12B (included below)

(A) Selective voxels for each preferred category are plotted on inflated views of individual subjects' brains. Negative selectivity values are not included on the brain maps because they obscure the preferred category-selective regions. (B) The distribution of selectivity values illustrates the roughly equal preference strength for food, faces, and places. Voxels showing selectivity for the category being measured are plotted in aqua, while voxels showing negative selectivity for that category are plotted in gold -- typically representing voxels in other category-selective regions.

We also comment on the shape and magnitude of these histograms in the text:

the pattern for each functional region is aligned with both the results from Experiment 1 and with prior results establishing selectivity for faces and places. Equally important, as illustrated in Figure~\ref{fig:selectivity-2}B, both the magnitude and distribution pattern of voxel-wise selectivity for food is on par with that for faces and places.

iii) the claim that face and food selectivity was similar based on those voxel maps is not quantified, and seems to be based on eyeballing the darkness of the red bits on figures S10 and S11. (If there is another method it should be made clear)

iv) The conclusion differs from the similar voxelwise analyses of the same data in Fig 6 in our paper:

https://web.mit.edu/bcs/nklab/media/pdfs/Khosla_CB2022.pdf

Of course it could be our quantification that was wrong :-) but I suspect not.

Given this concern the authors might consider alternative quantifications that enable the selectivity for food in the "food voxels" to be compared to selectivity for faces in the "face voxels".

We take the reviewer's point here to be to provide some quantitative assessment of the strength of selectivity for faces and food. The maps in Figures S11 and S12 do provide more information than the reviewer seems to give us credit for. Specifically, the number of voxels (using our more conservative measure of selectivity than was used in their work) that are food selective is certainly comparable to the number of face selective voxels - albeit with the constraint that selectivity for each category should only be prominent in the localized brain region associated with that category (which necessarily constrains the number of selective voxels). However, we do acknowledge that these maps do not provide any sense of the strength of such selective voxels (e.g., the face voxels could all be just barely have higher responses relative to the next highest responding category, while the food voxels might have much higher responses relative to the next highest responding category). To address this missing part of the picture, we now include histograms of the selectivity values for faces, food, and places for Experiment 2 in Figure S12B (see prior comment for additional information included in the manuscript discussing this figure). We view these histograms as directly addressing the reviewer's concern in that - as commented on by Reviewer #1 - the localizer uses "pure" food, face, and place images. The selectivity strengths in Experiment 1 - while qualitatively informative - are difficult to disentangle across categories because a single image might contain multiple categories, therefore reducing the difference between the nominally highest responding category and the next highest (e.g., faces vs. food is confounded because the face measure contains images of face+food and the food measure contains images of food+face).

REVIEWERS' COMMENTS:

Reviewer #1 (Remarks to the Author):

I've read the author's response and revised manuscript and appreciate their response to my questions and the discussion this generated. I'm not entirely surprised that a pure food PCA doesn't reveal immediately interpretable dimensions as I expect something more complex would be going on than simple semantic dimensions. A task for future studies to interpret.

In sum, despite all my quibbles with the PCA, I still think this is a great paper with two studies that mutually reinforce each other.

I have one final suggestion, though I DO NOT feel the paper needs to be held up for this if the authors disagree or if it's viewed as too complex. I've never been entirely comfortable with the tendency in fMRI to run PCA on concatenated datasets across subjects since it doesn't account for the dependencies in the data (i.e., each subject contributes 200 rows of data to the table). There are somewhat obscure methods that can perform PCA without concatenating or averaging subjects while still providing a PCA solution that best captures the group. Indeed, they even permit one to test whether that group solution is truly a good fit for all subjects, or whether subjects are too variable for a single PCA to describe their responses. The method I'm talking about is called STATIS (https://personal.utdallas.edu/~herve/abdi_Wires_AWVB2012_Final.pdf). Though it's a bit of a pain to implement and given that the norm in fMRI is to discard the subject dimension in these kinds of PCA analysis, I'm fine with that being the case here.